# What Is an "Arachnid"? Consensus, Consilience, and Confirmation Bias in the Phylogenetics of Chelicerata

**Prashant P. Sharma** [1,*] , **Jesús A. Ballesteros** [2] **and Carlos E. Santibáñez-López** [3]

1. Department of Integrative Biology, University of Wisconsin–Madison, Madison, WI 53706, USA
2. Department of Biology, Kean University, Union, NJ 07083, USA; jeballes@kean.edu
3. Department of Biology, Western Connecticut State University, Danbury, CT 06810, USA; santibanezlopezc@wcsu.edu
* Correspondence: prashant.sharma@wisc.edu

**Abstract:** The basal phylogeny of Chelicerata is one of the opaquest parts of the animal Tree of Life, defying resolution despite application of thousands of loci and millions of sites. At the forefront of the debate over chelicerate relationships is the monophyly of Arachnida, which has been refuted by most analyses of molecular sequence data. A number of phylogenomic datasets have suggested that Xiphosura (horseshoe crabs) are derived arachnids, refuting the traditional understanding of arachnid monophyly. This result is regarded as controversial, not least by paleontologists and morphologists, due to the widespread perception that arachnid monophyly is unambiguously supported by morphological data. Moreover, some molecular datasets have been able to recover arachnid monophyly, galvanizing the belief that any result that challenges arachnid monophyly is artefactual. Here, we explore the problems of distinguishing phylogenetic signal from noise through a series of in silico experiments, focusing on datasets that have recently supported arachnid monophyly. We assess the claim that filtering by saturation rate is a valid criterion for recovering Arachnida. We demonstrate that neither saturation rate, nor the ability to assemble a molecular phylogenetic dataset supporting a given outcome with maximal nodal support, is a guarantor of phylogenetic accuracy. Separately, we review empirical morphological phylogenetic datasets to examine characters supporting Arachnida and the downstream implication of a single colonization of terrestrial habitats. We show that morphological support of arachnid monophyly is contingent upon a small number of ambiguous or incorrectly coded characters, most of these tautologically linked to adaptation to terrestrial habitats.

**Keywords:** Arthropoda; circular reasoning; investigator bias; paleontology; phylogenomics

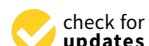

## 1. A Lesson from the Toucans

Around the time that we were Ph.D. students, one of the most memorable parables in science was passed down to us from Scott Edwards, a renowned ornithologist at the Museum of Comparative Zoology at Harvard University. At the very dawn of molecular phylogenetics in 1985, Scott, an undergraduate and aspiring junior scientist, ran into a colleague of his in the hallway of the Museum. His colleague, another student, proudly relayed, "I have found a synapomorphy for toucans!". The synapomorphy he was referring to pertained to a morphological character that united this particular group of birds, as evidence of their evolutionary relationship. Scott, to this day one of the most affable and collegial of evolutionary biologists, of course congratulated his fellow student. But he then mused: Did you encounter this synapomorphy because toucans are a natural group, *or because you went looking for it*?

Scott's coda addressed a broader question for phylogeneticists and scientists at large: How do we as natural historians distinguish observations of natural phenomena from investigator bias? That story, relayed to us 14 years ago at the time of this writing, deeply influenced the way we approach phylogenetic inquiry. It underscored the importance of

doubt in our outlook on the meaning of consilience in science, as well as how different phylogenetic data classes and competing topologies should be explored and evaluated.

That lesson bears heavily upon the phylogeny of Chelicerata, the subdivision of arthropods that includes groups like spiders, scorpions, and horseshoe crabs. In this review, we assess recent ideas and hypotheses pertaining to chelicerate relationships, with emphasis on the question of arachnid monophyly. We specifically scrutinize the practice of preferring only the topologies that are consistent with traditional, morphology-based relationships.

## 2. A Brief History of a Gordian Knot in Metazoan Phylogeny

The traditional understanding of chelicerate relationships is rooted in morphological data, which divide extant Chelicerata into three groups: Pycnogonida (sea spiders), Xiphosura (horseshoe crabs), and Arachnida (an assemblage of 12 terrestrial orders) [1–3]. Implicit in this topology is the hypothesis of a single colonization of land by the common ancestor of Arachnida. Within Euchelicerata (Xiphosura + Arachnida), extinct marine groups like Synziphosurina, Eurypterida, and Chasmataspidida (collectively thought to constitute a paraphyletic assemblage called "Merostomata") are thought to form a grade subtending Arachnida, reflecting a stepping-stone to the colonization of land by an aquatic ancestor [4–6]. Based upon an array of morphological characters, as well as their early appearance in Silurian and Devonian deposits, scorpions are generally reconstructed as the earliest (or one of the earliest) branching groups of arachnids by an array of historical and recent paleontological analyses [5,7–9] (Figure 1).

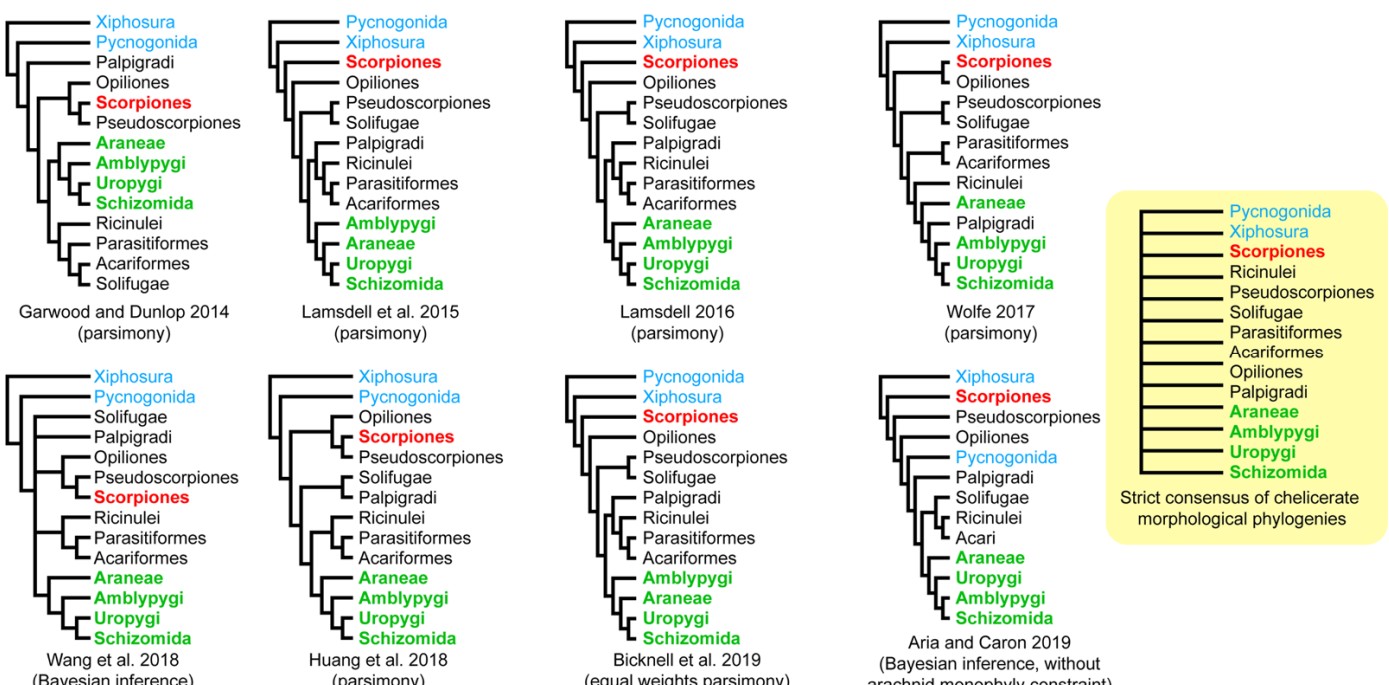

**Figure 1.** Phylogenetic relationships of chelicerate orders based on recent analyses of morphological datasets. Fossil taxa have been removed for clarity. Blue: marine orders; green: Tetrapulmonata; red: Scorpiones.

While the monophyly of Chelicerata and Euchelicerata (Xiphosura + Arachnida) has received broad support from analyses of morphological and molecular datasets, relationships within the arachnids remain unresolved, with the exception of a few key nodes (e.g., Arachnopulmonata sensu Ontano et al. [10]; Panscorpiones; Tetrapulmonata; Figure 1), which have largely been elucidated through phylogenomic datasets and rare genomic changes like whole genome duplication events. The fulcrum of the disagreement between morphological and molecular datasets comprises the monophyly of Arachnida itself. While



morphologists have typically considered arachnids to constitute a strongly supported monophyletic group, molecular sequence data have consistently struggled to recover this relationship (in some cases, even upon the inclusion of morphological data), since the first molecular phylogenetic datasets for Chelicerata were generated [11,12]. Sanger-sequenced datasets and analyses of mitochondrial genomes in particular could not recover arachnid monophyly, despite various approaches to phylogenetic analysis and inference [13–15]. This result was often dismissed on grounds of a wide array of putative artifacts, such as the aberrant morphology of the sea spiders, long branch attraction, and model misspecification. Lending credence to this argument was the clear evidence for heterogeneity in branch lengths across some arachnid taxa, with Acariformes, the non-opilioacariform Parasitiformes, and Pseudoscorpiones in particular exhibiting accelerated evolutionary rates that incurred the threat of long branch attraction artifacts [15–17].

The advent of genome-scale datasets facilitated a more precise examination of the source of this discordance. A 62-gene analysis of Regier et al. [16] was able to recover arachnid monophyly in only two of four analyses, and with limited support (68% and 80% bootstrap support). The authors of the work interpreted this to mean that Arachnida was "strongly recovered", though other relationships of comparable depth were recovered more consistently and with higher support values (typically, maximal nodal support values across all four analyses). Subsequently, a 3644-gene analysis of Sharma et al. [17] discovered a faint signal supporting arachnid monophyly in slowly evolving genes. It was shown that the 500- and 600-slowest evolving genes in the matrix, when concatenated, could recover the monophyly of Arachnida with maximal nodal support, though this signal vanished upon addition of faster evolving genes.

This study was often mistakenly interpreted to mean that slowly evolving genes were more accurate than faster-evolving genes in the face of long branch attraction, reflecting the underlying assumption that arachnids must be monophyletic. However, this conclusion is contradicted by the existence of localized peaks of nodal support—comparable to that of Arachnida—for mutually exclusive hypotheses also supported by slowly-evolving genes (e.g., the sister group relationships of pseudoscorpions to either scorpions or Acari exhibit localized peaks of support in rate-subsampled matrices [17] (Figure 2a). Therefore, the ability to find local peaks of nodal support within an ordered subsampling of genes when long branch attraction is incident cannot be a guarantor of phylogenetic accuracy. Indeed, subsequent works based on phylogenetically informed orthology criteria for gene selection could not replicate this signal for Arachnida (Figure 2b). Sharma et al. [17] articulated the concern that searching for arachnid monophyly may represent an idiosyncratic goal and that the traditional interpretation of a single terrestrialization event was undermined by the robust and derived placement of scorpions as the sister group of the tetrapulmonates. This group was named Arachnopulmonata [17], a clade of the only extant arachnid orders that bear book lungs.

The significance of the scorpion placement is twofold. First, paleontologists had previously inferred several Paleozoic stem-group scorpion fossils to be marine or at least aquatic, based on the fossil assemblages where they were discovered and the existence of putative gills in the mesosoma of these extinct groups. Under this scenario, a derived placement of scorpions would imply multiple terrestrialization events, or a return to aquatic habitat in the branch subtending the modern scorpions (more recently, given the proliferation of support for Arachnopulmonata, paleontologists have subsequently revised this interpretation to suggest that all Paleozoic scorpions must have been terrestrial, the presence of gilled fossils like *Waeringoscorpio* notwithstanding; see Howard et al. [18]). Second, the derived placement of scorpions proximal to tetrapulmonates suggested a derived origin of the book lung—a respiratory organ thought to represent the internalized counterpart of the horseshoe crab book gill [19]. The robust recovery of Arachnopulmonata in phylogenomic datasets was further substantiated by a series of rare genomic changes stemming from the discovery of a shared whole genome duplication event in the common ancestor of the arachnopulmonates [20–24]. Paralleling the history of the waves of genome

duplication in the vertebrates, genomes of Arachnopulmonata were shown to exhibit two Hox clusters [23], broad retention and transcriptional activity of duplicated paralogs of developmental patterning genes [22], enrichment of specific microRNA families [21], and arachnopulmonate-specific divergence of gene expression patterns of duplicated paralogs [25–27]. Additional datasets from recently established tetrapulmonate model systems (e.g., tarantula; whip spiders [28–31]), as well as non-arachnopulmonate outgroups (e.g., mite; tick; harvestman; [32–34]) further buttressed this inference, and additionally revealed that the fast-evolving pseudoscorpions definitively constitute a member of Arachnopulmonata [10]. Intriguingly, extant Xiphosura exhibit a shared two- or possibly three-fold genome duplication on the branch subtending the four living species, but these events are unrelated to the arachnopulmonate duplication [35–38].

Due to the systemic nature of whole genome duplication events, the weight of evidence was decidedly in favor of phylogenomic results and contrary to morphological analyses (Figure 2c). This is because the incidence of shared duplications results in specific and replicated gene tree topologies, as well as divergent gene expression patterns ensuing paralogs, across hundreds of retained duplicates; explaining these genomic phenomena as the result of independent events becomes implausibly non-parsimonious. As a result, in the span of a few years, decades-old hypotheses based on morphology—such as scorpions constituting the sister group of harvestmen [2,39] or the remaining arachnids [1,40], or pseudoscorpions the sister group of solifuges [1,2]—were refuted, now considered the likely consequences of morphological convergence and ensuing misinterpretations of similar feeding structures and shared arrangements of respiratory organs (e.g., two-segmented chelate chelicerae and tracheal arrangement in pseudoscorpions and solifuges; anatomy of the book lungs and book gills of scorpions and merostomates, respectively; preoral chambers formed from gnathobases in scorpions and harvestmen [2,19,39,40]).

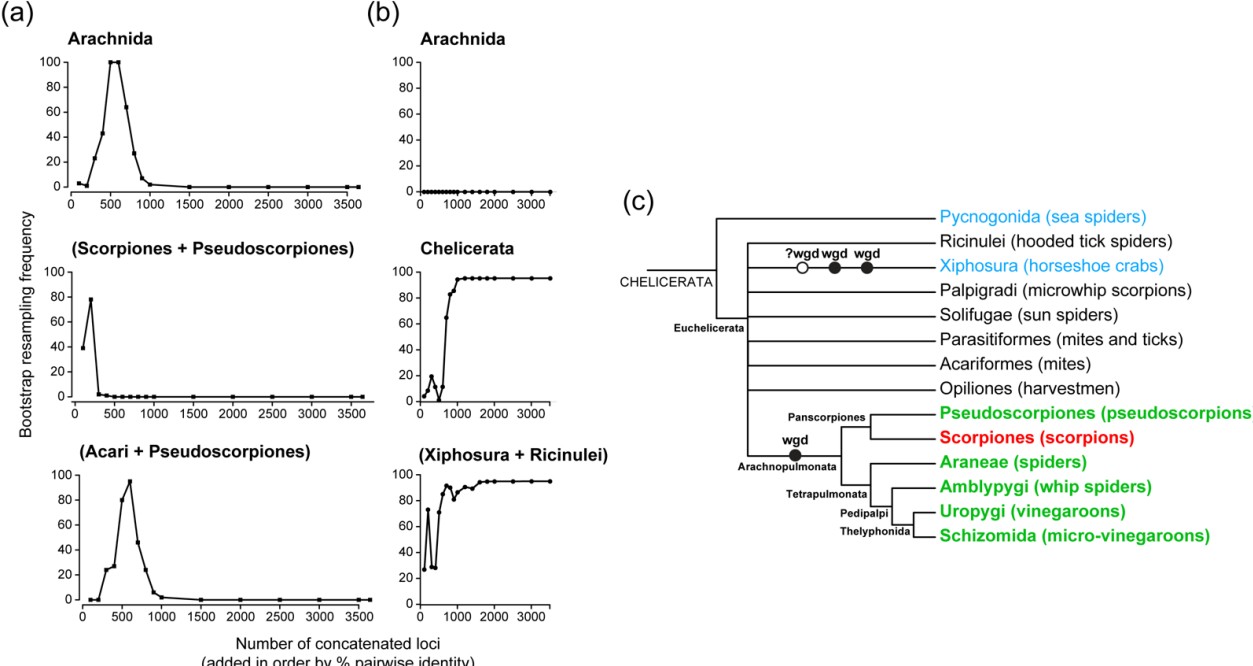

**Figure 2.** The ability to find localized peaks of support for nodes in slowly evolving genes is not a guarantor of phylogenetic accuracy. (**a**) Selected nodal support trajectories from Sharma et al. [17]. The bottom pair of nodes are mutually exclusive, and the bottom-most node (Acari + Pseudoscorpiones) has been falsified by rare genomic changes [10]. (**b**) Selected nodal support trajectories from Ballesteros and Sharma [41]; no support was recovered for Arachnida even among slowly evolving genes. The support trajectory for a nested placement of Xiphosura (bottom) is closely comparable to that of a robustly supported node (Chelicerata; middle). (**c**) Simplified phylogeny of Chelicerata showing inferred locations of whole genome duplications (WGD). Note that no morphological phylogeny has ever recovered this composition of Arachnopulmonata. Colors in (**c**) follow Figure 1.

### 3. The Debate over Arachnid Monophyly

A more definitive debate about arachnid monophyly in molecular datasets was addressed in the works of Ballesteros and Sharma [41] and Lozano-Fernández et al. [42]. The former work came to the conclusion that there was no support for arachnid monophyly. Ballesteros and Sharma [41] implemented analyses of gene-wise log-likelihood score (ΔGLS) to dissect phylogenetic signal; simulations of incomplete lineage sorting to assess explanatory vehicles for arachnid non-monophyly; and multiple approaches to species tree inference. Their orthology inference approach made use of phylogenetically unrooted topologies, which has been shown to outperform BLAST-based or distance-based calculations of orthologs [43]. Of the 106 analyses performed, the only analysis that could recover Arachnida was one where ΔGLS was used to identify and concatenate a minority of genes (33.6%) that supported this relationship. This minority of genes showed no evidence of being "better" or more accurate than the majority, with respect to an analysis of 70 metrics for systematic bias, such as saturation, evolutionary rate, or missing data (Figure 3 of Ballesteros and Sharma [41]). They were also able to show that the nested placement of Xiphosura was not attributable to long branch attraction; upon removing all long orders from the analysis in taxon deletion experiments, they still found Xiphosura as derived within the arachnids (Figure 5 of Ballesteros and Sharma [41]). Moreover, they were able to show that detection and isolation of genes supporting specific relationships with ΔGLS could be used to recover nonsensical relationships, such as a grouping of chelicerates with Pancrustacea, with maximal nodal support (Figure 8 of Ballesteros and Sharma [41]). This thought experiment served to reinforce that the ability to find matrices supporting specific relationships with maximal support is not synonymous with phylogenetic accuracy, as cherry-picking genes that support preconceived hypotheses is a circular exercise.

These lessons were ignored by the subsequent work of Lozano-Fernández et al. [42] who analyzed chelicerate phylogeny with a larger sampling of taxa. This work analyzed three matrices using maximum likelihood and Bayesian inference approaches. One of these (Matrix A) was composed using preselected genes, whose origins and basis for selection are not reproducible (ref. [44]). The second (Matrix B) was the result of a distance-based algorithmic approach to orthology inference (OMA; [45]). The third (Matrix C) was the result of uniting the first two matrices, after excluding duplicates. They found that one of their three matrices (Matrix A) was able to recover arachnid monophyly, though only after the removal of six taxa from the analysis, and only using the site heterogeneous CAT+GTR+Γ model implemented by PhyloBayes-mpi [46], a computationally intensive Bayesian inference approach. The same approach, when applied to their other matrices, failed to recover arachnid monophyly. To understand why, the authors examined saturation plots of each matrix (a metric for multiple substitutions at the same sites; a correlate of evolutionary rate) and observed that Matrix A had a slightly better value for saturation (a slope of 0.38) in comparison to the other two matrices (0.33 for both Matrices B and C). Lozano-Fernández et al. [42] reached the conclusion that concomitant application of denser taxonomic sampling, the use of site heterogeneous models, and filtering for unsaturated genes could recover Arachnida (as well as Acari), and therefore, phylogenetic accuracy. The ability to recover arachnid and acarine monophyly in one out of seven analyses was touted as an example of "consilience" in phylogenetics.

Notably, both of these previous studies [41,42] were missing a handful of key lineages—namely, the miniaturized arachnid orders Palpigradi and Schizomida, and the rare, slowly evolving parasitiform lineage Opilioacariformes [15,47]. A subsequent investigation of chelicerate phylogeny by Ballesteros et al. [44] was able to include these taxa for the first time in a combined phylogenomic framework. This work was similarly unable to recover arachnid monophyly, despite the application of site heterogeneous models developed for maximum likelihood frameworks, as well as through the use of PhyloBayes-mpi. To understand why, Ballesteros et al. [44] added these three taxa to Matrices A and B of Lozano-Fernández et al. [42] and reran analyses using both maximum likelihood and

PhyloBayes-mpi. The results were the same—Ballesteros et al. [44] could not recover arachnid monophyly using this approach either.

In the course of those analyses, Ballesteros et al. [44] discovered a series of grave analytical and bioinformatic errors in the works of Lozano-Fernández et al. [42], which were detailed in a supplementary document (Supplementary Text S2 of Ballesteros et al. [44]). These included errors in the filtering of loci by taxonomic completeness; incorrect measurement of per-locus saturation; incorrect calculation of linear regressions for reporting saturation due to a well-known error in a previous version of the software Microsoft Excel; inconsistency in the definition of saturation across studies by the same research team (compare the calculation of slope in [42] versus [48]); and lack of convergence of results generated using PhyloBayes-mpi. While the Lozano-Fernández et al. research team attempted to address these issues in a follow-up study using a derivation of Matrix B (by Howard et al. [18]), this work was similarly shown to suffer from additional bioinformatic errors, with some genes appearing more than once in the principal Howard et al. matrix (ref. reanalyses by Ontano et al. [10]). In addition, Howard et al. [18] preselected another pair of historical datasets (two that were known to yield arachnid monophyly under certain substitution models) and analyzed all datasets using the CAT-Poisson model in PhyloBayes-mpi. Upon recovering support for arachnid monophyly in all applications of CAT-Poisson, they reached the conclusion that these results substantiated the accuracy of Arachnida.

This result is outright contradicted in at least one case (the 500-slowest evolving genes matrix of Sharma et al. [17]), due to (1) the demonstrable analytical superiority of the CAT-GTR+Γ [49] model (explicitly deemed the best model and the one preferred across works by Lozano et al. [42,48]) over the CAT-Poisson model, and (2) a previous analysis of this same matrix using CAT-GTR+Γ by Sharma et al. [17], which recovered arachnid paraphyly with maximal nodal support (Figure 7 of Sharma et al. [17]). Howard et al. [18] also ignored as part of their reanalyses any matrices built using slowly evolving genes that did not recover arachnid monophyly, and for which previous analyses using CAT-GTR+Γ also refuted arachnid monophyly [41,44].

Putting aside the error-prone phylogenetic analyses, flexible criteria for dataset diagnosis, and willingness to dismiss contradictory evidence from the literature, there are two major concerns with the conclusions drawn by Lozano-Fernández et al. [42] and Howard et al. [18].

First, the addition of just two to three terminals to the matrices of both studies (which achieves complete sampling of all extant chelicerate orders), as well as a key parasitiform lineage (Palpigradi, Schizomida, and Opilioacariformes, respectively), consistently results in the disruption of both arachnid and acarine monophyly with support, even when these matrices were reanalyzed using identical algorithmic approaches and substitution models [10,44]. These reanalyses suggest that the monophyly of Arachnida and Acari requires the exclusion of certain lineages that undermine these traditional groupings.

Second, the matrices that were able to recover these groups are rife with paralogs; 29% (68/233) of loci in the Lozano et al. [42] Matrix A and 41% (82/200) of loci in the Howard et al. [18] matrix were detected as including clear paralogs, using an annotation approach based on the *Drosophila melanogaster* proteome [50]. This discovery suggests that the monophyly of Arachnida may be an artifact reflecting noise and bioinformatic error, rather than phylogenetic signal.

Despite the flaws of the Lozano-Fernández et al. [42] and Howard et al. [18] studies, these works, and the attendant conclusion of arachnid monophyly, continue to receive widespread support among morphologists and paleontologists. Adherents of arachnid monophyly seize upon the existence of molecular matrices that can recover arachnid monophyly with maximal support (as well as the claim that these partitions are less saturated or somehow more accurate), citing these as either: (1) outright evidence for the accuracy of arachnid monophyly [42]; or (2) evidence that molecular data are at least partly congruent with arachnid monophyly (or are ambiguous on the matter), which is argued

should trigger a deference to the traditional morphological understanding of chelicerate relationships [18]. The implicit argument of the latter claim is steeped in the assumption that morphology is a reliable arbiter of deep phylogenetic relationships in Chelicerata.

We put both of these claims to the test.

## 4. How to Get the Tree You Want without Really Trying

Lisa: "By your logic I could claim that this rock keeps tigers away."

Homer: "Oh, how does it work?"

Lisa: "It doesn't work."

Homer: "Uh-huh."

Lisa: "It's just a stupid rock."

Homer: "Uh-huh."

Lisa: "But I don't see any tigers around, do you?"

Homer: "Lisa, I want to buy your rock."

—*The Simpsons*, 1996

For phylogenomic datasets, it is generally understood that nodal support in the form of resampling techniques and posterior probabilities is not a reliable measure of the underlying signal in concatenation-based phylogenetic approaches. Maximal support values are commonly obtained in phylogenomic datasets above a certain size, but these may be attributable to amplification of noise, rather than signal. Exploration of datasets and dissection of systematic biases at the level of genes and sites is critical to understanding which nodes exhibit inter-partition conflict or lack of phylogenetic signal. Rigorous investigations of phylogenetic signal within supermatrices are especially crucial when long branch attraction artifacts are incident, as these tend to drive high nodal support values in cases of model misspecification or undersampling of fast-evolving lineages.

As for saturation, minimizing bias resulting from saturated sites not a controversial goal, but approaches to mitigating saturation are mixed in efficacy for Chelicerata. A partial solution is to restrict analyses to slowly evolving genes (as saturation is correlated with evolutionary rate; but see Figure 2b). Solutions based on the recoding of datasets (e.g., Dayhoff recoding) have retrieved uninformative results, due to the abrogation of signal at the base of Euchelicerata [42,44]. Site heterogeneous models are similarly variable as potential solutions for recovering Arachnida (discussed above). For this review, we used the same contextual definition of saturation (unusually, measured for a supermatrix, rather than individual genes) used by Lozano-Fernández et al. [42].

To explore the reliability of nodal support and saturation as metrics for phylogenetic accuracy, we undertook a series of thought experiments originally proposed by Ballesteros and Sharma [41], using as our source data Matrices A and B of Lozano-Fernández et al. [42]. We specifically explored the possibility of discovering partitions that support alternative groupings, with comparison of saturation rates for these matrices versus the original values of Matrices A and B.

Using the approach of Shen et al. [51], we dissected support for four debunked hypotheses of arthropod relationships: Cormogonida (the sister group relationship of Pycnogonida to the remaining arthropods); Atelocerata (Myriapoda + Hexapoda); Schizoramia (Crustacea + Chelicerata); and Myriochelata (Myriapoda + Chelicerata). To discover partitions supporting each grouping, we compared ΔSLS distributions for unconstrained trees versus trees constrained to recover each of these older groupings, computing both sets of maximum likelihood trees under site heterogeneous (PMSF) models, following approaches previously detailed by us [41]. We concatenated partitions supporting each grouping and inferred the resulting maximum likelihood trees using a site heterogeneous model. Nodal support was inferred using ultrafast bootstrap resampling. As shown in Figure 3, we were able to construct matrices from both Matrix A or Matrix B that could

recover each of these groups, with nodal support (>95% bootstrap) for all four groupings in one or both matrices.

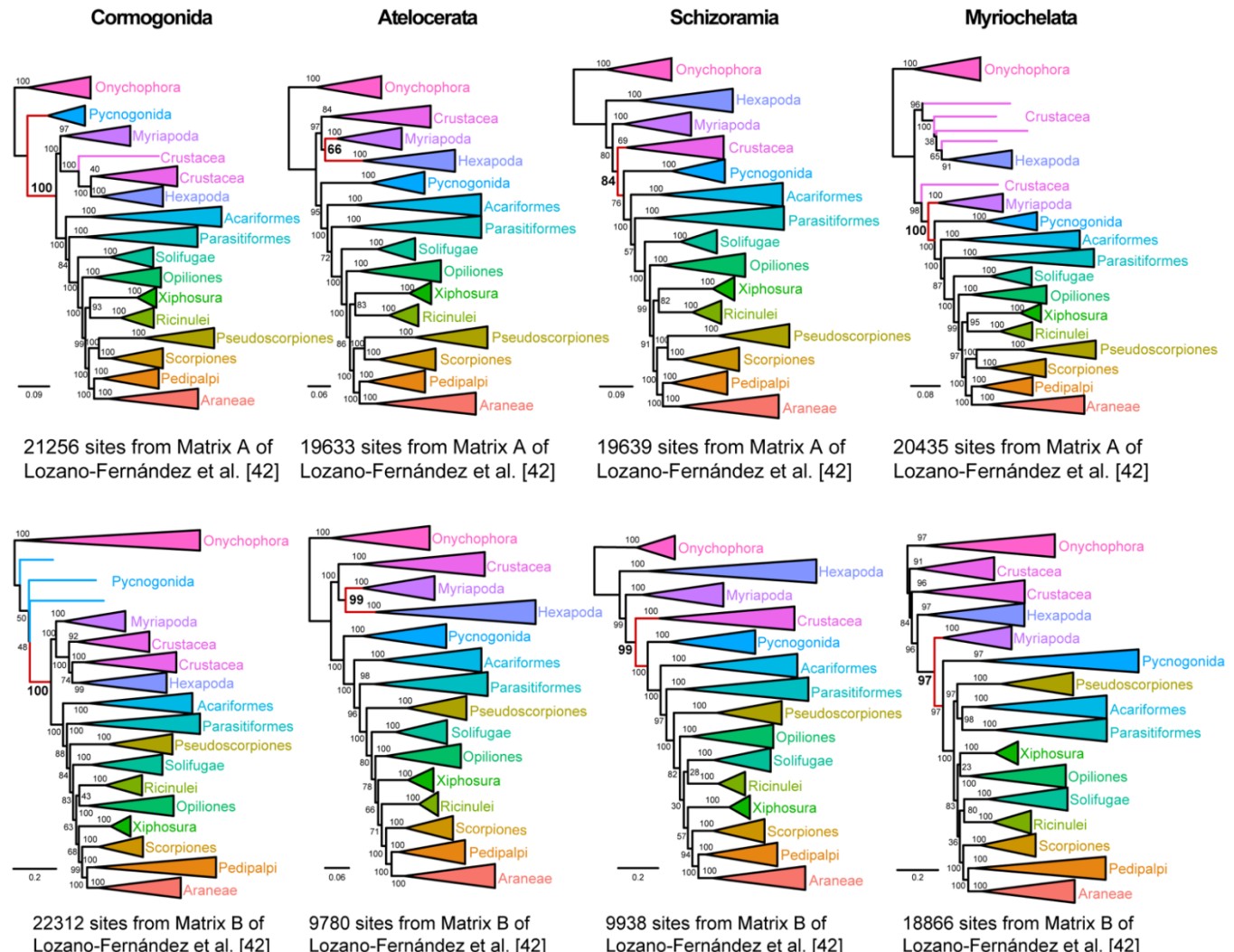

**Figure 3.** Cherry-picking molecular data to recover preconceived, traditional relationships can be used to justify debunked groupings. Top row: Topologies derived from Matrix A of Lozano-Fernández et al. [42]. Bottom row: Topologies derived from Matrix B of Lozano-Fernández et al. [42].

To those uninitiated in arthropod phylogeny, the ability to generate such data matrices may be construed as a lack of clear signal in basal arthropod relationships. Could such matrices imply a hidden signal for traditional groupings that were once supported by certain subsets of morphological characters? To dispel this notion, we proceeded to generate three additional matrices that supported completely nonsensical groupings: a clade of Xiphosura and Crustacea, to the exclusion of hexapods and other chelicerates (inspired by the whimsical notion of "making horseshoe crabs crabs again"); a clade of scorpions and spiders, to the exclusion of the other tetrapulmonates (the arachnids that most frighten people); and a clade of Pycnogonida + *Drosophila melanogaster*, to the exclusion of all other hexapods and chelicerates (taxa studied by Thomas Hunt Morgan; it is a little-known fact that the earliest works of the father of the first arthropod model organism addressed the development of sea spiders [52]). As shown in Figure 4, we were just as able to easily construct matrices from both Matrix A and Matrix B that could recover absurd groupings, with nodal support (>90% ultrafast bootstrap) in one or both analyses.

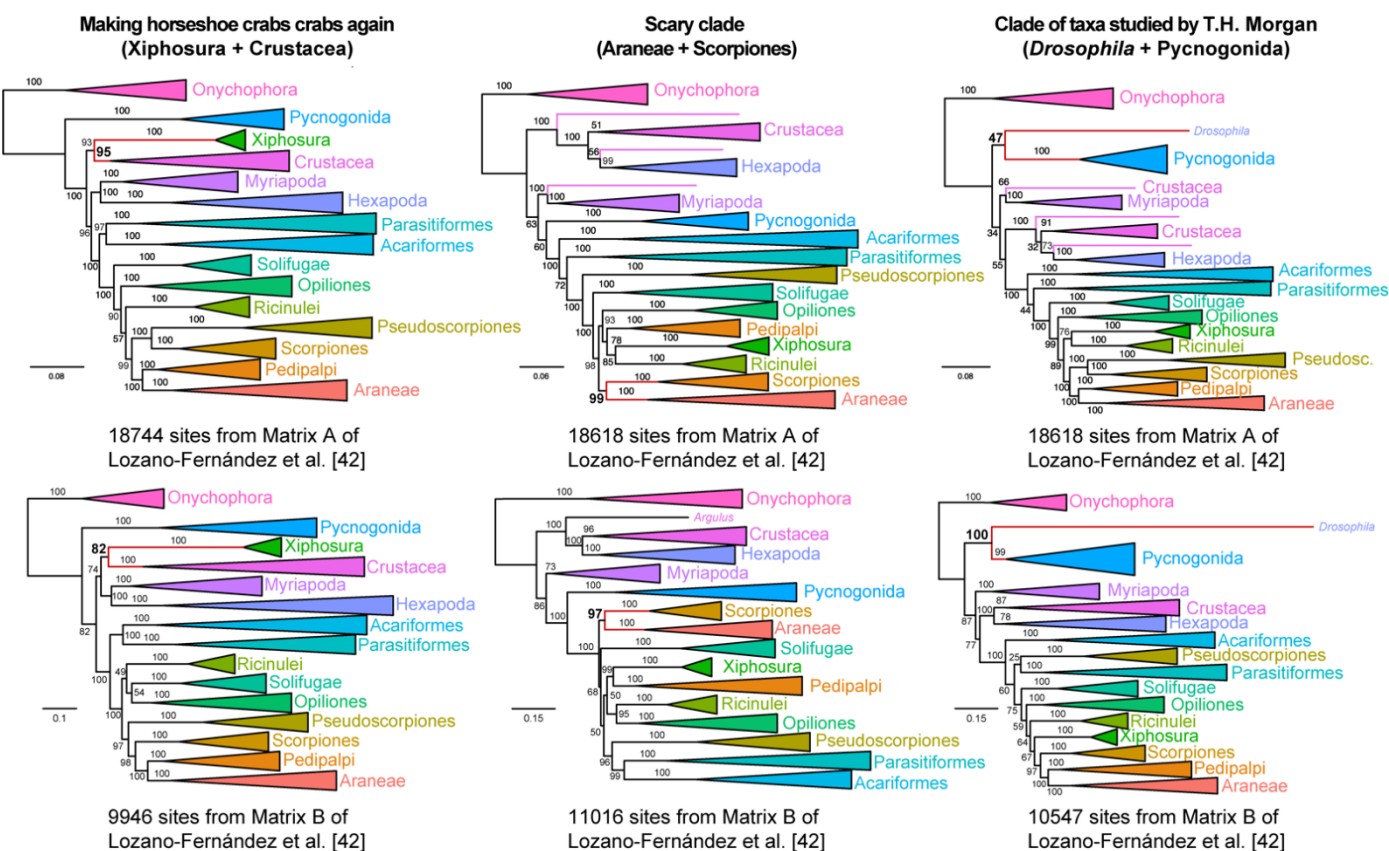

**Figure 4.** Taken to an extreme, cherry-picking molecular data can recover completely nonsensical relationships, often with support. Top row: Topologies derived from Matrix A of Lozano-Fernández et al. [42]. Bottom row: Topologies derived from Matrix B of Lozano-Fernández et al. [42].

Surely, one might think these matrices must compare poorly to actual phylogenetic matrices. We would expect such datasets to be smaller than unconstrained subsets or exhibit aberrant values for various measures of systematic bias, such as saturation. To test this, we generated values for saturation for every dataset generated from genes in Matrices A and B, to compare these to the original values reported in the study of Lozano-Fernández et al. [42]. Our approach to measuring saturation was identical to that implemented by Lozano-Fernández et al. [42], with the exception that we did not use an erroneous version of Excel to calculate coefficients of correlation (correctly measured, all $R^{22}$ values exceeded 97%). As shown in Figure 5, every Matrix A-derived dataset recovering spurious relationships exhibited equal or better saturation values than Matrix A; and three Matrix B-derived datasets recovering absurd groupings outperformed Matrix B.

These analyses underscore that the ability to generate a matrix that can recover a preconceived result with maximal support does not equate with phylogenetic accuracy. They further reinforce the broadly understood principle that looking for a post hoc justification that validates a specific outcome in science (in the case of Lozano-Fernández et al., a metric for saturation, which they have defined inconsistently from one study to the next [42,48]) is the epitome of confirmation bias. In the specific case of saturation, it is well known that this metric is not a strong predictor of phylogenetic accuracy by itself, as elegantly demonstrated by Mongiardino Koch [53].

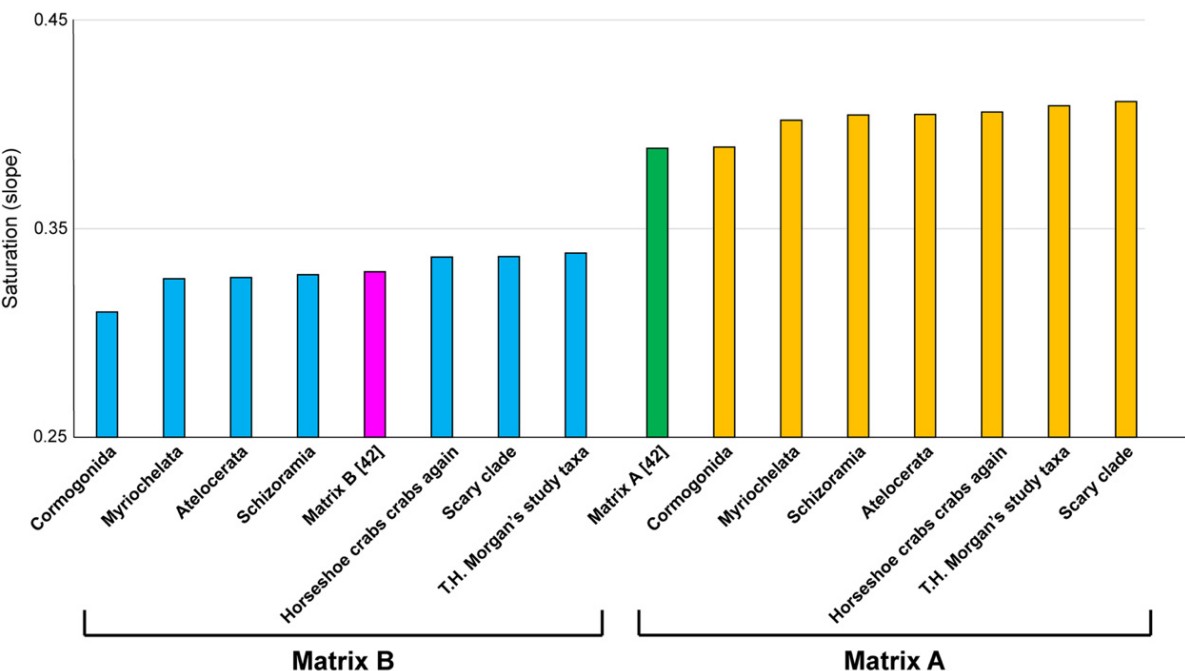

**Figure 5.** The saturation value of a supermatrix is not a valid proxy for phylogenetic accuracy (contra [42,48]). Slopes measured for matrices supporting debunked or absurd groupings can outperform Matrices A and B of Lozano-Fernández et al. [42], when drawn from the same underlying genes. (In other words, if you fell for this argument, as articulated by Lozano-Fernández et al. [42] in support of arachnid monophyly, you got fooled.).

This bears directly upon the epistemology of Lozano-Fernández et al. [42] and Howard et al. [18], whose interpretation of phylogenetic accuracy is predicated on the ability to recover a preconceived, traditional hypothesis (which is circular). Notably, Howard et al. [18] used the same approach as Sharma et al. [17] with respect to approach to orthology inference and the selection of slowly-evolving genes as the filtering criterion—they unwittingly evaluated the same set of orthogroups that had already been analyzed by the earlier work, but cast their 200-locus dataset as independent verification; in reality, the overlap between the gene sets is so extensive that the Howard et al. [18] 200-locus matrix cannot be treated as an independent analysis, as much as a recapitulation of a published result. Moreover, neither Lozano-Fernández et al. [42] nor Howard et al. [18] ever directly addressed the observation that other workers were unable to recover arachnid monophyly when restricting analyses to slowly-evolving or less saturated genes in other datasets, nor using site heterogeneous models (including CAT-GTR+Γ in PhyloBayes-mpi) on such filtered datasets [41,44]—the silver bullets they proposed for consistent recovery of arachnid monophyly across datasets do not work. It is particularly damning that expanding taxonomic sampling to sample all extant chelicerate orders in their datasets, and thereafter reanalyzing those datasets using identical methods, only serves to destabilize this node, with support [10,44]. This rejection of evidence that contradicts a preconceived or preferred hypothesis, as well as the unwillingness to test a broader range of datasets more rigorously and dispassionately, may reflect a calcified bias by adherents of "molecular paleobiology" in support of traditional, morphology-based relationships.

Parenthetically, "molecular paleobiology" refers to a loosely defined school of systematics that aims to reconcile paleontological and molecular phylogenies, ostensibly placing priority on the morphological and paleontological data partitions (as exemplified by the works in question [18,42]). Note that here we distinguish between molecular paleontology in the strict sense (e.g., the study of fossilized biomarkers; sequencing of ancient DNA) and "molecular paleobiology" in the sense of a specific phylogenetic research program (hence, we use quotation marks here to distinguish this approach to phylogenetics). "Molecular paleobiologists" champion the integration of morphological and molecular datasets as

the best approach to phylogenetics. Most of the activities of this school are no different from what molecular phylogeneticists simply call molecular dating, which is a widely implemented technique. Some of the recurring controversies engendered by "molecular paleobiologists" have stemmed from overconfidence in the completeness of the fossil record, overconfidence in morphological phylogenies (or specific, arbitrarily selected morphological phylogenies, when competing hypotheses are available), overvaluing morphological and paleontological evidence in the assessment of competing molecular hypotheses, overinterpreting paleontological and morphological data in establishing calibrations in molecular dating; outright misconstruing or misrepresenting the morphological literature to accord with a preferred molecular topology; and consistently ignoring the repeated observation by other research groups that site heterogeneous models (as well as other proposed silver bullets to achieving morphological trees with molecular data) do not actually achieve the desired result across datasets [10,44,54–59]. However, given that morphological hypotheses of relationships predate molecular counterparts by decades or sometimes centuries, "molecular paleobiological" results are often unguardedly accepted by the broader community for their palatability, specifically by those who do not examine the underlying phylogenomic data and analyses or lack the expertise to do so.

Returning to the matter of chelicerate relationships, the most recent work addressing these issues comprehensively sampled all extant chelicerate orders with a 506-taxon phylotranscriptomic study, in tandem with site heterogeneous models (including with PhyloBayes-mpi implementation) and investigations of the claim that genes supporting arachnid monophyly were less biased or "better" at inferring deep relationships. Ballesteros et al. [50] showed that rich sampling of Chelicerata only further undermined support for arachnid and acarine monophyly. Filtering for less saturated genes and use of site heterogeneous models (PMSF and CAT-GTR+Γ models) not only refuted arachnid monophyly, but also revealed that the monophyly of Acari was a long branch artifact, driven by the exclusion of slowly-evolving parasitiform taxa like Opilioacariformes in the analyses of Lozano-Fernández et al. [42] and Howard et al. [18] (see also reanalyses by Ballesteros et al. [44] and Ontano et al. [10] on the effect of including Opilioacariformes to their matrices). Ballesteros et al. [50] also dismantled the unsubstantiated notion that genes supporting the nested placement of Xiphosura exhibited artifacts; to the contrary, they were able to show that genes supporting arachnid monophyly, which were consistently in the minority across datasets, tended to be short and bear few parsimony-informative sites. Short genes with few informative sites have been closely linked to systematic artifacts and poor phylogenetic signal. Apropos, upon examining the distribution of signal across sites, they showed that sites supporting arachnid monophyly exhibited high levels of Shannon entropy, reflecting noise rather than signal. Tellingly, the number of sites supporting debunked, artificial groupings (e.g., Dromopoda, an erstwhile grouping of scorpions, pseudoscorpions, solifuges, and harvestmen, which has been refuted by rare genomic changes) exceeded the number of sites supporting Arachnida.

Simply stated, unbiased analyses of molecular data do not support arachnid monophyly. The ability to contrive matrices that can do so through cherry-picking of genes and taxa, as well as finding post hoc justifications for said matrices, is the modern equivalent of hunting for synapomorphies for preconceived groups. We submit that cherry-picking datasets and trees should be treated with the same level of skepticism as the obsolete practice of single-character systematics. In the best light, preferring only those trees that confirm morphology-based hypotheses of phylogeny reflects a form of confirmation bias that stems from a naïve misunderstanding of how signal and noise are distributed in molecular data at the genomic scale. In the worst light, this practice represents a gateway to pseudoscience.

## 5. Morphology in the Era of Phylogenomics

"He rain-made you. A guy says if you pay him, he can make it rain. You pay him. If and when it rains, he takes the credit. If and when it doesn't, he finds reasons for you to pay him more."

—Maurice Levy, *The Wire*, 2004

"The morphological data show … very few nodes … have significant resampling support and the strict consensus is poorly resolved. This is unquestionably a limitation of the taxon to character ratio used in the present study … The character sample could potentially be bolstered studies [*sic*] on other character systems … "

—Ganske et al., 2021 [60]

Before we address morphological support (or the lack thereof) for Arachnida, we consider here the broader role and value of morphological datasets in modern phylogenetics, with a critical eye toward the future of morphology in a phylogenomic era.

Over the course of the past 30 years, morphological data have declined in relevance as data sources for phylogenetics, and a paucity of research groups continue to examine both morphological and molecular datasets toward the goal of empirical systematics (e.g., [15,60–66]). Part of the reason for this, as diplomatically articulated by Giribet [67], is tied to declining costs of sequencing, as well as the difficulty of articulating clear homology statements for problematic morphological character systems. As a fuller (and blunter) answer, the decline of morphology's prominence in modern phylogenetics has just as much to do with the multifaceted superiority of molecular data as predictors of phylogeny. While these concepts below are fairly well known (if tacitly acknowledged) by the broader community of evolutionary biologists, we posit them here as a prelude to the debate over arachnid monophyly.

Firstly, molecular data present a universal character system. Molecular matrices contain an alphabet that is universally applicable to all cellular life, and thus any researcher can take a molecular matrix and work with it, for any group of organisms, with no barriers or steep learning curves for understanding the underlying data. Genomic sequencing can be performed with standardized approaches that are broadly transferable to all taxonomic groups. By contrast, understanding, interpreting, and analyzing morphological character systems requires familiarity with the morphological characters and states that underlie the matrix. This means that the data contained in a given morphological matrix can only be understood by taxonomic experts of that group (in some cases, a mere handful of individuals), barring a steep learning curve for novices to that taxon. At its core, a morphological character matrix is the product of one researcher (or team) coding subjective interpretations, often with little understanding of character dependencies or underlying costs of state transformations. For this reason, two different morphologists can (and routinely do) come up with markedly different interpretations, matrices, and character states for the same taxa. This makes the detection and correction of coding errors in historical morphological matrices more difficult (discussed below) and hinders the integration of different sets of morphological matrices with non-overlapping characters or character states.

Second, molecular sequences present the desirable quality of scalability. Molecular matrices greatly exceed morphological datasets in size and variance of evolutionary rates, particularly with respect to different parts of a genome and nucleotide versus peptide sequences. This means that molecular data can just as easily inform population genetics and microevolutionary processes, as the deep phylogenetic relationships and the origins of major taxonomic groups. By contrast, the informativeness of morphological data occupies a middle ground between the two extremes. Morphological characters do not evolve rapidly enough to inform population-level processes, which makes them irrelevant for such disciplines as population genetics and epidemiology (there are several good reasons why you do not see epidemiologists tracking the spread of COVID-19 using analyses of viral morphological trait matrices). At the same time, practicing phylogeneticists have broadly

suspended efforts to create morphological datasets spanning pre-Paleozoic divergences, due to the paucity of available character systems and clear homology statements. As examples, the Metazoan Tree of Life team abandoned, after enormous investment of time and expense, efforts to establish the homology of characters that could apply to all extant animal phyla, because defining homologies at this scale had proven intractable (G. Giribet, personal communication). By the same token, constructing morphological datasets spanning even older lineages (Opisthokonta; Eukarya; the entire tree of life) is simply not feasible and our understanding of the basal-most splits in the Tree of Life today is predominantly informed by molecular sequence data.

Molecular sequence data can be collected inexpensively, reliably, reproducibly, and expeditiously. The collection of morphological data has remained precisely as painstaking, laborious, and slow as it was in the 1980s. While innovations like micro-CT scanning have improved the recovery and resolution of internal morphological characters, the coding of morphological matrices is simply not scalable to the level of genomic sequencing. The length of the operation aside, morphological data (from micro-CT or traditional microscopy) need to be interpreted and coded, which requires time, labor, and lineage-specific expertise. The coding itself demands questionable practices like subjective discretization of character states and arbitrary treatments of phenomena like phenotypic plasticity and polymorphism.

Thirdly, morphological data matrices have consistently struggled with the problem of character independence. Morphological characters are effectively black boxes, whose independence can be approximated only using congruence-based approaches (i.e., comparisons of state changes between characters on a tree), which is inherently circular. Molecular sequence data, in tandem with complete genomes, provide a straightforward and reliable way to assess the independence of loci, as a function of their distributions across the genome; in principle, character independence of loci can be quantitatively characterized.

A fourth consideration pertains to models of evolution. Due to the proliferation of molecular sequence data in the last half-century, substitution models available for both nucleotide and peptide alignments have become increasingly sophisticated, being informed by statistical and biochemical validation. This has enabled such applications as molecular divergence dating, assessment of compositionally heterogeneous evolution, assessment of heterotachous evolution, and measurement of incomplete lineage sorting. Even in cases where molecular phylogenies have initially faltered (as was the case of Myriochelata in arthropod phylogeny or the non-monophyly of Protobranchia in Sanger datasets of bivalves [68,69]), additional molecular sequence data, rare genomic changes, and/or improved substitution models accounting for asymmetrical branch lengths have facilitated the identification of the artifacts driving those results [70–72]. As stated above, morphological characters remain to this day a black box, rife with both theoretical and empirical pitfalls [73,74]. We lack reasonable models for their evolution, which has precipitated marked controversy over how morphological data matrices should be analyzed [75–80]. As a result, even in the context of molecular dating, morphological data have come to play a supporting role to molecular data in modern phylogenetics.

All these considerations bear upon the utility and accuracy of relationships predicted by morphological data matrices. Scores of systematic works have rejected or refuted traditional morphology-based relationships in the light of molecular data and subsequent reappraisals of morphological homology statements (e.g., Articulata; Aschelminthes; Coelenterata; Coelomata; Polychaeta). This is particularly the case for deeper nodes in the Tree of Life, groups that lack a large number of available character systems for scoring, and the higher-level relationships of groups prone to morphological convergence. As a result, most modern approaches to understanding the evolution of morphology in extant taxa will generate a molecular phylogeny first and map morphological character states onto the molecular tree thereafter. In the case of conflicts between morphological and molecular trees, the latter usually turn out to be more accurate, having prompted a reevaluation of morphological homology statements and character definitions. Prominent examples of

these trends include the systematic history of Ecdysozoa, Pancrustacea, Gnathifera, and Ambulacraria [81–85].

Even in the case of groups with well-established morphological datasets and fairly clear evolutionary history, evidence for the superiority of molecular data is commonplace. In the case of Bivalvia, Bieler et al. [61] assessed the informativeness of different morphological character systems under a statistical framework, comparing the result to a total evidence tree of nine genes and morphology. They were able to show that more than 50% of characters coded exhibited phylogenetic signal indistinguishable from random structure, for all character systems except for external shell morphology and sperm ultrastructure. Upon extracting the 99 characters that exhibited any signal at all and computing a tree (Figure 35 of Bieler et al. [61]), they were able to recover the monophyly of only two of the six major bivalve lineages—a result that can be surpassed with a dataset of just four nuclear coding genes with a fraction of the effort, time, and expense required for collecting morphological data [70]. At a shallower phylogenetic node, Zou and Zhang [86] examined patterns of homoplasy in a mammal dataset of 3414 parsimony informative morphological characters and 5722 parsimony informative amino acid sites. They were able to show that morphological data were more prone to convergence than amino acid sites, in large part due to the small number of character states defined for discretized morphological characters. Similarly, in the case of biogeography, a recent meta-analysis of morphological and molecular phylogenies showed that molecular phylogenies exhibit better congruence to biogeographic distributions than their counterpart morphological trees [87], suggesting that homoplasy in morphological trees obscures inference of macroevolutionary processes.

The logical conclusion stemming from these trends should be that morphological data are non-universal and often unreliable arbiters of deep phylogenetic relationships, and therefore constitute an inferior data class for numerous taxa. Yet few published works arrive at this explicit conclusion. The archetypal conclusions postulated by phylogenetic workers that compare data classes to the detriment of morphology are (1) the importance of fossils and an integrated understanding of evolutionary history, which substantiates the continued relevance of morphology (e.g., [67,87,88]); and/or (2) a call for collecting more morphological data, despite the molecular data usually having resolved most of the relationships in question more efficiently and more reproducibly (e.g., [60]). Given the comparatively greater epistemological and practical challenges to collecting, interpreting, and analyzing morphological data in the context of phylogenetic inference, it is not surprising that morphological datasets have taken on an ancillary role by comparison to molecular data, in deciphering evolutionary relationships of extant taxa.

The considerations above bear directly upon the evaluation of morphological evidence in support of arachnid monophyly.

## 6. Morphological "Support" for Arachnida Is Grounded in Errors, Shared Absences, Homoplasy, and Circular Reasoning

> "Promoting or defending a specific phylogenetic hypothesis via lists of compatible synapomorphies is a common but problematic approach ... A node supported by a long list of synapomorphies may seem convincing taken in isolation but may become less acceptable when its full phylogenetic implications are explored."
>
> —Jeffrey W. Shultz (2007) [2]

The base of Euchelicerata constitutes a recalcitrant node where molecular data do not yield clear answers, whereas morphological data are perceived to support arachnid monophyly unambiguously. But how strong is the actual evidence for this relationship? To examine this question, we revisited a series of morphological matrices recently produced by paleontologists and reanalyzed them using identical approaches as reported in their respective publications. We mapped onto the resulting trees all characters that constitute unambiguous synapomorphies for Arachnida. We additionally tabulated puta-

tive synapomorphies for Arachnida from historically significant works from the literature (Table 1).

We excluded from this analysis one dataset that was never published by the author [89] and another that could not recover arachnid monophyly without enforcing a topological constraint for this preconceived relationship [9].

For two datasets that we analyzed, one lacked a published character list accompanying the morphological matrix [90], and the second—a well-known and oft-cited work touting the reconciliation of morphology and molecules through the consideration of fossils—published an incorrect version of the morphological matrix with no correspondence to its character list [91]. Repeated requests to the author of this series of matrices [89–91] over the past two years for the correct versions of datasets elicited a bizarre online exchange that led us to conclude that the original matrices were lost or incorrectly analyzed. Documentation of this exchange is available upon request. The versions that were eventually and sporadically supplied to us did not match their published counterparts with respect to the number of taxa and characters analyzed. The author of the aforementioned matrices openly communicated to us their awareness of these issues over the past several years but has taken no steps to correct any of them. We are baffled as to how some researchers and editors in the field of arthropod paleontological systematics are able to operate with such lax standards and permissive requirements for science; for the rest of us, such oversights would be tantamount to trolling for journal retractions.

In any case, the series of matrices provided to us by the aforementioned author [89–91], as well as one that was correctly published [92], analyzed with or without a character list, recovered no characters constituting unambiguous synapomorphies for Arachnida.

Next, we examined a family of matrices emphasizing the sampling of aquatic euchelicerates (the merostomates) [5,8,93]. The character and taxon set for these studies exhibited broad overlap and all three recovered the same two characters as unreversed synapomorphies of Arachnida: (1) the identity of the first podomere of the fifth prosomal appendage that protrudes beyond the carapace, and (2) the condition of exopod armature, other than lamellae.

The first of these characters was coded as the second podomere (the trochanter) for all arachnids, a higher value for merostomate taxa (which exhibit comparatively greater lateral outgrowth of the carapace margin), and inapplicable for Pycnogonida (as sea spiders lack a carapace). The coding of this character is incorrect. In Palpigradi, Acariformes, and Solifugae, the carapace does not include the segments of the third or the fourth walking leg. In addition, in Solifugae and Palpigradi, the protruding coxae of these two leg-bearing segments are clearly visible in dorsal view; these arachnid orders should rightfully have been coded as inapplicable for this character, like the sea spiders (Figure 6a,b). Moreover, the designation of this character is arbitrary; comparable characters do not exist for any other prosomal appendages except for the fifth prosomal appendage in these matrices, suggesting that this character was erected for the express purpose of artificially distinguishing arachnids as a natural group.

The second character (exopod armature other than lamellae) is also incorrectly coded. It was scored as absent for all arachnids, present (setae or spines) for merostomates, and inapplicable for sea spiders. This coding is not logical; if coded as inapplicable in sea spiders because they lack exopods, then this character should also be coded as inapplicable in arachnid orders that unambiguously lack exopods as well (i.e., the non-arachnopulmonate arachnid orders). Alternatively, it should be coded as absent in sea spiders, following the same logic as the coding for arachnids. A broader issue with such character codings, recapitulating Ballesteros et al. [50], is that they group taxa based on shared absences that may be the result of convergence (especially in terrestrial habitats; see also [74]). Assuming for the sake of simplicity that this character was coded correctly (with only arachnids sharing a state of absence), it implies that the simplification or loss of exopods reflects a homologous condition. As revealed by the evolutionary history of hexapods, myriapods, and other terrestrial pancrustacean groups (e.g., terrestrial amphipods and

isopods), the reduction or total loss of exites, rendering an arthropod limb uniramous, has evolved repeatedly, likely as a result of convergent adaptations to terrestrial habitats. Given the evolutionary history of the mandibulate limb in terrestrial taxa, the lack of exopods or exopod armature in terrestrial chelicerates does not offer compelling evidence uniting Arachnida.

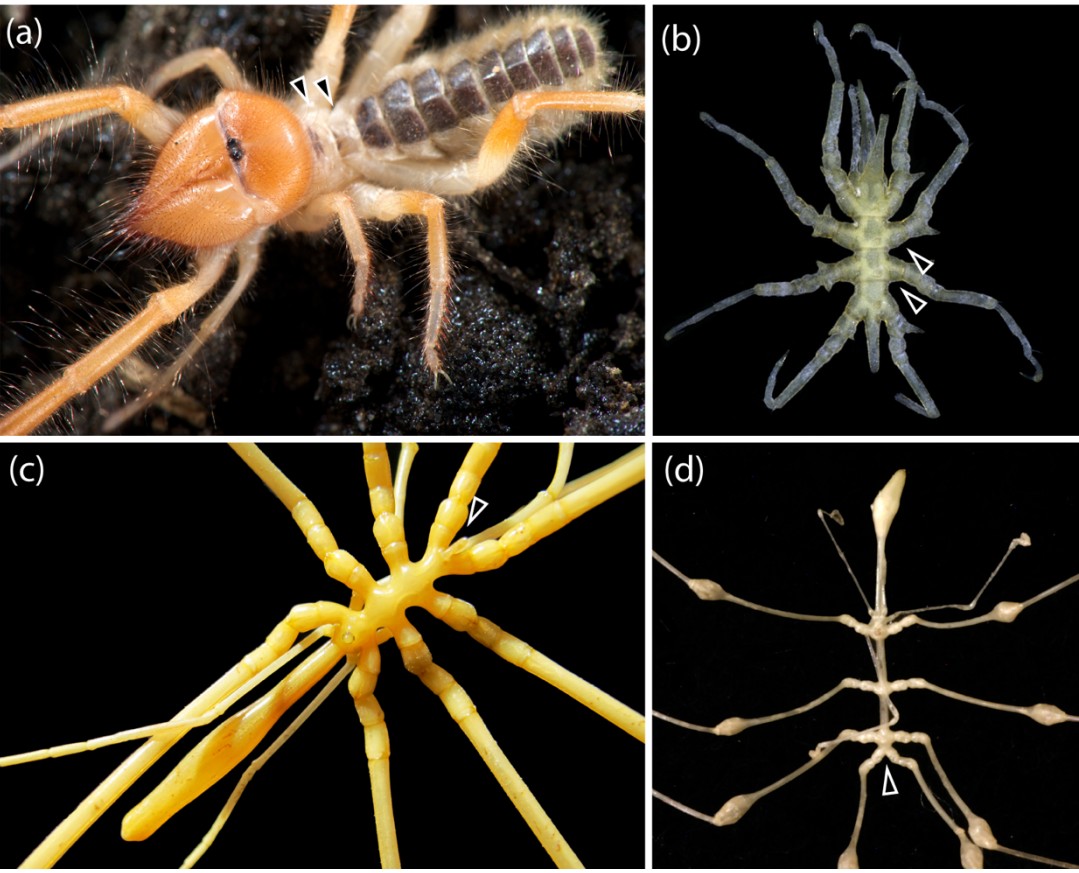

**Figure 6.** Incorrect codings of morphological character states underlie the recovery of Arachnida in morphological matrices. (**a**) Live habitus of Solifugae (*Eremobates* sp.). Arrowheads show the locations of the third and fourth walking leg segments, which are not part of the carapace (the protruding coxae are clearly visible from a dorsal view). The same condition is found in Palpigradi. (**b**) Micro-CT of an austrodecid sea spider (*Austrodecus glaciale*). Arrowheads indicate the visible coxae of walking leg segments. (**c**) Unsegmented and appendage-free anal tubercle (the remnant of the opisthosoma; arrowhead) of a colossendeid sea spider (*Colossendeis megalonyx*). (**d**) Dorsal view of the colossendeid *Rhopalorhynchus magdalena*. In such species, the opisthosomal rudiment has been lost altogether (arrowhead). Note as well the absence of exopods in sea spiders. Photographs: (**a**) G. Giribet; (**b**) G. Brenneis; (**c**,**d**) C.P. Arango.

The last series of morphological datasets we analyzed was drawn from a family of matrices originally generated by Shultz [2,39], who rooted Arachnida using Xiphosura, presuming arachnid monophyly. We analyzed derivations of this matrix, which subsequently underwent adaptations and expansions to include sea spiders and other phylogenetically significant lineages [94–97]. Of the morphological matrices we examined in the recent literature, this family of matrices represents the most intensive sampling of extant and fossil arachnids. In this family of matrices, we recovered the same two synapomorphies for Arachnida previously reported by Garwood and Dunlop [94]: (1) presence/absence of the appendages of the first opisthosomal segment (absent in arachnids; present in all other chelicerates), and (2) origin of the walking leg apotele depressor (tibial in arachnids; tarsal in horseshoe crabs and sea spiders).

**Table 1.** There is no compelling morphological support for the monophyly of Arachnida. Tabulation and critical evaluation of putative synapomorphies historically uniting Arachnida.

| Source; Year | Character | Putative State Shared by Arachnida | Evaluation |
|---|---|---|---|
| Weygoldt and Paulus [1]; 1979 | extraintestinal digestion | present | Absent in Opiliones, various Acariformes, and various Parasitiformes |
| | Malpighian tubules | present | Absent in Palpigradi, Pseudoscorpiones, Opiliones, and some groups of Acariformes |
| | eyes | disintegrated into five or fewer ocelli | Semi-aggregate eyes present in fossils of Scorpiones, Ricinulei, and Trigonotarbida |
| | slit sensilla | present | Absent in Palpigradi |
| Shultz [98]; 2001 | reduced pleural fold (doublure) in the prosomal carapace | present | Also shared by Pycnogonida |
| | slit sensilla | present | Absent in Palpigradi |
| | anterodorsal rotation of anterior prosoma resulting in anteroventrally directed mouth | present | Same as the ancestral condition in the common ancestor of Pycnogonida |
| | appendages on somite VII (O1) in adults | absent | Shared absence; comparable to absence of tritocerebral appendages in hexapods and myriapods |
| | cardiac lobe or glabella on carapace | absent | Shared absence |
| | medial genital opening (paired in Xiphosura) | single | Initially paired in embryonic arachnids; if only scoring adults, character does not apply in Pycnogonida, so paired gonopores would be synapomorphic for merostomates |
| | appendages on somite XIII | absent | Shared absence if correctly scored; in fact, derivatives of O2 coxae form the genital operculum |
| | postcerebral crop and proventriculus | lost/reduced | Shared absence |
| | anterior oblique axial muscles | absent | Shared absence |
| | attachments of opisthosomal posterior oblique axial muscles | pleural | Cannot be polarized because sea spiders lack this character; tergal attachments in Xiphosura could represent a synapomorphy of the merostomates |
| | endosternal suspensors of somites I and II | absent/detached | Shared absence |
| Shultz [2]; 2007 | carapacal pleural doublure | absent | Shared absence |
| | cardiac lobe | absent | Shared absence |
| | pedal gnathobases | absent | Shared absence |
| | moveable endites | absent | Shared absence |
| | aerial respiration | present | Prone to generating suites of morphological homoplasies (e.g., Atelocerata; Pulmonata); tautological |
| | anteriorly or anteroventrally directed mouth | present | Same as the ancestral condition in the common ancestor of Pycnogonida |

**Table 1.** *Cont.*

| Source; Year | Character | Putative State Shared by Arachnida | Evaluation |
|---|---|---|---|
| Legg et al. [91]; 2013 | none | none | Note: Incorrectly published online; original version lost |
| Legg [89]; 2014 | none | none | Note: Matrix never published; analyzed version provided to us mismatches publication |
| Briggs et al. [92]; 2016 | none | none | Note: Character list not published |
| Siveter et al. [92]; 2017 | none | none | |
| Lamsdell et al. [93]; 2015 | first podomere of pa5 that fully projects beyond carapace | second podomere | Incorrectly scored for Solifugae, Palpigradi, Acariformes |
| | exopod armature other than lamellae | absent | Incorrectly scored for Pycnogonida |
| Lamsdell [5]; 2016 | first podomere of pa5 that fully projects beyond carapace | second podomere | Incorrectly scored for Solifugae, Palpigradi, Acariformes |
| | exopod armature other than lamellae | absent | Incorrectly scored for Pycnogonida |
| Bicknell et al. [8]; 2019 | first podomere of pa5 that fully projects beyond carapace | second podomere | Incorrectly scored for Solifugae, Palpigradi, Acariformes |
| | exopod armature other than lamellae | absent | Incorrectly scored for Pycnogonida |
| Garwood and Dunlop [94]; 2014 | appendages on opishosomal segment 1 | absent | Incorrectly scored for Pycnogonida |
| | origin of apotele depressor | tibia | Ambiguous to interpret for Xiphosura and Pycnogonida due to variation in podomere counts |
| Huang et al. [97]; 2018 | appendages on opishosomal segment 1 | absent | Incorrectly scored for Pycnogonida |
| | origin of apotele depressor | tibia | Ambiguous to interpret for Xiphosura and Pycnogonida due to variation in podomere counts |

　　　　Again, both characters are ambiguously coded. For the former, sea spiders were scored as having appendages on the first opisthosomal segment—a problematic designation, given that sea spiders do not have an opisthosoma, but rather, an abdominal rudiment (at most) that is unsegmented and does not bear appendages (Figure 6c,d). The only fossil sea spider known to bear a segmented opisthosoma (*Flagellopantopus blocki*) also shows no appendages on this posterior tagma. Why then were sea spiders coded as bearing opisthosomal appendages?

　　　　Garwood and Dunlop [94] argued that the alignment of sea spider and arachnid body segments implies the positional homology of the first opisthosomal segment of arachnids and the fourth walking leg segment of sea spiders, due to the presence of an additional oviger-bearing segment in the sea spider cephalon. The problem with this coding is that it reflects an internal inconsistency across their matrices. Within chelicerates, the number of prosomal segments is generally fixed, but it exhibits additions in sea spiders (at least three origins of 10- and 12-legged genera [99,100]) and reductions or losses in groups like eriophyoid mites (four-legged upon hatching, due to early attenuation of the L3 and L4 segments during embryogenesis) [101]. In none of the sea spider genera with supernumerary appendages is the opisthosoma lost or otherwise affected, suggesting that the addition

of prosomal segments does not require altering the boundary or identity of prosoma and opisthosoma (similarly, reduction of the posterior prosomal segments in eriophyoid mites does not impact the opisthosomal boundary). Opisthosomal segmentation is far more variable in extant chelicerates (e.g., between two and 13 segments across extant arachnids; unsegmented in sea spiders). The ensuing inconsistency is that other characters in the same matrix were coded as reflecting the absence of an opisthosoma in sea spiders altogether (e.g., characters pertaining to the condition of opisthosomal dorsal and ventral segmentation were coded as inapplicable), which conflicts with the definition of the fourth walking leg segment as "opisthosomal". A less tortuous coding would list the condition of the first opisthosomal appendage as absent in sea spiders, which renders this character as an invalid synapomorphy of arachnids.

The character of the apotele depressor stemmed from a comprehensive work by Shultz [98] on the musculoskeletal anatomy of *Limulus polyphemus*, wherein Shultz defined a series of putative synapomorphies of the arachnids (drawing upon his previous seminal work in 1990 [39]). Homologizing the podomeres and muscle attachment sites in extant chelicerates is complicated by the absence of the metatarsus in horseshoe crabs and the presence of additional podomeres in sea spiders—the same segmental landmarks do not exist in arachnid versus other chelicerate walking legs. Moreover, as previously argued by Ballesteros et al. [50], the use of this entire character system as a source of arachnid synapomorphies may be confounded by convergent evolution in terrestrial habitats, comparable to convergences in the musculoskeletal system of hexapods and myriapods. Arthropod appendages are highly adaptable structures that have undergone markedly different selective pressures in terrestrial versus aquatic environments, resulting in parallel evolution of numerous podomeres and rami [102].

An anemic defense of arachnid monophyly was recently mounted by Howard et al. [18] on morphological grounds. In their "holistic" treatment of chelicerate relationships, Howard et al. did not present a single morphological analysis, restricting their analytical contributions to a paralogy-riddled molecular analysis (discussed above; see also Figures S2 and S3 of Ontano et al. [10] and associated discussion), in addition to a discursive overview of a handful of morphological character systems in the context of arachnid evolution, such as the anatomy of the lateral eyes, the respiratory organs, and the feeding mouthparts. As previously rebutted by Ballesteros et al. [50], every one of the morphological character systems discussed by Howard et al. [18] has been shown to exhibit misleading levels of homoplasy in the phylogeny of Mandibulata, with many of these character systems contributing to the flawed traditional interpretation that hexapods and myriapods were sister groups (Atelocerata), representing a single colonization of land. The discourse of Howard et al. [18] consistently fails to grasp the implication of the repeated failure of morphological and paleontological datasets to recover the only higher-level chelicerate relationships that have been robustly supported by multiple data classes (i.e., Arachnopulmonata sensu Ontano et al., Panscorpiones; [10]): If morphological datasets are unable to recover these splits (all of which were accepted as valid by Howard et al. [18], and solely on the basis of molecular support), then other nodes recovered by morphological analyses of Chelicerata could just as easily be inaccurate as well.

This fundamental contradiction in reasoning aside, Howard et al. [18] cursorily pointed to the historical availability of "substantial lists of morphological autapomorphies . . . proposed to support Arachnida" to promote the perception of unambiguous support for this grouping. Ironically, the 11 synapomorphies they refer to were previously defined in 2001 by Shultz [98], who later disparaged the practice of listing synapomorphies and character recycling [2]. As shown in Table 1, six of these 11 synapomorphies are shared absences, which do not constitute compelling apomorphies uniting any group (consider, for example, the patterns of shared absences observed in hexapods and myriapods for biramous appendages, gills, and tritocerebral appendages). Two others are predicating on ignoring character states in Pycnogonida that are shared with the arachnids (i.e., they reflect the practice of conceptualizing a tree of Arachnida rooted with Xiphosura a priori

and dismissing the morphology of sea spiders as aberrant [2]). One character cannot be polarized due to the morphology of sea spiders and may in fact constitute a synapomorphy of merostomates. Another two characters still are in fact not arachnid synapomorphies at all but are restricted to subsets of terrestrial orders or to specific developmental stages. Shultz himself heavily revised the list of potential arachnid synapomorphies by 2007 and reduced it to six [2], recognizing that characters like slit sensilla and fluid feeding must have evolved within a subset of the arachnids (but see Table 1 for critical evaluation and rebuttals of all six characters). Even characters that clearly reflect functional adaptations to terrestrial habitats, like Malpighian tubules, are restricted to subsets of arachnid orders. Certain complex characters, like Malpighian tubules and tracheal tubules, are known to have evolved repeatedly across Panarthropoda, undercutting the value of excretory and respiratory organs as homoplasy-free character systems.

In reality, the "substantial lists" of arachnid synapomorphies alluded to by Howard et al. [18] do not exist, nor does the ability to conjure such characters (even if they existed) forfend the possibility of phylogenetic inaccuracy—consider, for example, the longer putative lists of synapomorphies once invoked to support the erstwhile groupings Articulata (Arthropoda + Annelida) and Atelocerata (Hexapoda + Myriapoda). There is little uniting Arachnida except for the condition of being mostly terrestrial and shared absences that likely stem from being terrestrial. Efforts to seek out support for this relationship with the goal of "rescuing morphology" risk incurring further circularity and confirmation bias.

To arachnologists—as well as to paleontologists with actual expertise in arachnids (e.g., [4,94])—none of this is new or surprising, given the discoveries and advances in arachnid biology in the past twenty years. Practicing arachnologists have understood for decades that morphological data are not strongly dispositive with respect to several competing hypotheses of various interordinal relationships (e.g., the placement of Opiliones, Palpigradi, and Pseudoscorpiones [15,44,103–105]), with various character systems demonstrably prone to convergent evolution at both deep and shallow taxonomic scales (e.g., respiratory organs) [10,106]. As a recent example, Ballesteros et al. [50] recently generated a comprehensive morphological matrix sampling 514 fossil and extant chelicerates, which they analyzed using both parsimony and Bayesian inference approaches. Neither analysis supported arachnid monophyly, with the base of Euchelicerata constituting a soft polytomy. When the morphological matrix was paired with their 506-taxon phylogenomic dataset, Merostomata was resurrected as a clade derived within Arachnida, with support. This total evidence analysis is the only case where morphological data, analyzed alone or in combination with molecules, have ever been able to recover relationships supported by rare genomic changes (Arachnopulmonata and Panscorpiones [10]). A more proximal placement of Merostomata to Arachnopulmonata may reconcile both the evolution of trabeculae in a derived group of chelicerates [6], as well as century-old observations of morphological correspondences between merostomates and scorpions—a stepwise colonization of land may indeed apply, but only for one derived group of arachnid orders..

To rule out the possibility that biased coding of the morphological matrix had driven this result, Ballesteros et al. [50] additionally paired their molecular dataset with two recent morphological matrices generated by paleontologists, with widely differing taxon sets. The inclusion of molecular data dissolved much of the higher-level structure of chelicerate relationships, but in all cases consistently refuted arachnid monophyly. In the case of a comprehensively sampled morphological matrix of Panarthropoda, the inclusion of molecular data again recovered Merostomata as nested within the arachnids as well. These analyses suggest that morphological support for traditional chelicerate relationships is not as robust as typically perceived, in the specific context of total evidence.

The perception that Arachnida is well-supported by morphology likely reflects a century-old recycling of an antiquated idea, predicated on the historical belief that terrestrialization was rare or irreversible in evolutionary history. This belief is shown to be groundless when examining the modern (i.e., molecular) phylogeny of Mandibulata.

Putting aside the myriapod and hexapod terrestrialization events, the history of decapod terrestrialization has been shown to result from at least ten colonizations of land, and a recent molecular work has even revealed the diphyly of the terrestrial isopods [107,108]. Thus, even on relatively shallow timescales, terrestrialization is common across Arthropoda, and a clear driver of morphological convergence, which obscures phylogenetic signal in morphological datasets. Similarly, a complex history of terrestrialization with repeated colonization of land has been revealed in other phyla, such as Nematoda and Mollusca (at least 12 colonizations of land in "Pulmonata"), through the lens of phylogenomics [109,110]. Arachnida is little more than the newest member of a growing list of erstwhile specious terrestrial groupings that reflects cases of multiple terrestrialization events.

## 7. Arthropod Systematic Paleontology and the Value of Validation

"Ever bought a fake picture, Toby? . . . The more you pay for it, the less inclined you are to doubt its authenticity."

—George Smiley, *Tinker Tailor Soldier Spy*

"There has been growing research interest on how people respond to corrections of misinformation . . . This body of research has converged on the conclusion that corrections are rarely fully effective: that is, despite being corrected, and despite acknowledging the correction, people by and large continue to rely at least partially on information they know to be false. This phenomenon is known as the continued-influence effect . . . In some circumstances, when the correction challenges people's worldviews, belief in false information may ironically even increase . . . "

—Stephan Lewandowsky, Ullrich K.H. Ecker, John Cook (2017) [111]

In his eulogy to the waning prominence of morphological data in 21st century phylogenetics, Giribet [67] articulated two broad reasons for maintaining research programs in morphology: an understanding of morphological evolution for its own sake, and molecular dating (with emphasis on total evidence dating, or tip-dating). Similar arguments were expounded by Lee and Palci [88], who extolled the availability of morphological characters as a source of validation for molecular clades (increasingly, a role superseded by analyses of genomic architecture and rare genomic changes), as well as their essential function in molecular dating. Indeed, in cases of taxa with (1) a richly detailed fossil record and (2) strong morphological signal in hard parts that are prone to fossilization, signals from morphological and molecular data partitions have been shown to complement one another and successfully yield an integrated dated resolution of extinct and extant lineages (e.g., [61,64,112]). Using taxon addition and deletion experiments for an array of morphological matrices, Mongiardino Koch and Parry [113] recently showed that fossil taxa can have profound impacts on tree topology, usually more so than extant taxa. Notably, however, Panarthropoda was one of two test cases where neontological data had greater impact than fossil data [113]. This may be because the authors of that Panarthropoda matrix coded molecular data, such as Hox gene expression, as morphological characters for extant taxa—a possibility we cannot test, given that the authors published an incorrect matrix version with a mismatching character list [91].

Overall, such works [67,88,113] make a compelling argument for the consideration of fossil taxa both for improved tree searches and for a more comprehensive understanding of character state evolution. In principle, the inclusion of fossil taxa in phylogenies toward a more comprehensive understanding of evolutionary history is not controversial and represents a desirable goal. The benefits of adding taxa to matrices is broadly recognized, particularly in the context of breaking up long-branch attraction artifacts (e.g., Acari; Pseudoscorpiones + Acariformes). If >95% of species have undergone extinction during the Phanerozoic, surely a comprehensive view of phylogenetic history must account for character states and histories of these groups. Would the integration of morphology and fossils not help to resolve the mystery of chelicerate relationships?

Putting aside the practical and algorithmic challenges of analyzing such combined datasets (specifically, the relative weights assigned to morphological versus molecular partitions), what proponents of the total evidence approach often overlook is that their favored test cases generally tend to be taxa that bear an exceptionally detailed, highly complete fossil record, with strong morphological signal in the body parts that fossilize in those groups. Examples of taxa that meet these criteria include the echinoderms, shelled mollusks, and various groups of vertebrates [61,62,112]. However, for many soft-bodied and/or small-bodied invertebrate lineages (e.g., Platyhelminthes; Placozoa; Cycliophora), the fossil record is prohibitively sparse for total evidence approaches, due to a lack of hard parts prone to fossilization. Most terrestrial lineages of animals (e.g., the arthropods) tend to have poorer fossil records than aquatic counterparts, save for cases of exceptional preservation (but for a counterexample, consider the fossil record of Pycnogonida [4,100]). As a result, empirical total evidence matrices for such groups tend to be flooded with missing entries for fossil taxa, which can result in the destabilization of the tree topology upon the inclusion of fossils. Among Chelicerata, the degree of missing data and uninformative characters for orders like Phalangiotarbida results in these groups acting as rogue taxa in Bayesian inference analyses [50,94,96].

Worse still, in some groups of invertebrates, external morphological characters that are observable in fossils may not be those that retain high phylogenetic signal. Such is the case for many groups of chelicerates. As examples, in Pycnogonida, it was recognized fairly early that the traditional morphological interpretation of a reductive trend in appendages across the sea spider tree was not substantiated by phylogenetic analysis. Subsequent comparisons of morphological and molecular data (either in isolation or in total evidence analyses) revealed marked incongruence in topology, partly attributable to the low number of codable characters in the sea spider bauplan, in addition to homoplasy [99,114]. In Solifugae, higher-level taxonomy is diagnosable only using a subset of feeding appendage characters, and in many cases, the diagnostic character states are only available in adult males. For Scorpiones, a higher-level phylogenomic analysis showed that external morphological characters were either uninformative or incongruent with higher level relationships inferred by thousands of genes [115]. Only a subset of internal morphological characters pertaining to embryonic developmental mode, the distribution of the digestive glands, and the morphology of the hemispermatophores is consistent with the molecular phylogeny and informative at higher taxonomic levels [115,116]. As with various arthropod taxa, numerous scorpion groupings based on traditional morphological analyses were shown to be non-monophyletic, requiring extensive systematic revision of higher-level relationships through the lens of phylogenomics [115,117–119]. Even at shallow taxonomic scales, parametric analyses of shape data have shown that structures traditionally held to harbor high phylogenetic information (e.g., carapace or pedipalpal podomere shape) exhibit extensive morphological convergence or broadly uninformative variation within derived scorpion groups [120]. Internal characters are virtually inaccessible to paleontologists, with the rare exceptions of some arachnid fossils (e.g., Opiliones) with exquisite preservation of intromittent organs [63,121].

These trends are uniquely problematic for chelicerate paleontologists, for whom fossilized traits available for coding occur in a small sample of specimens and constitute mostly external traits prone to morphological stasis or low phylogenetic signal. But limitations of morphological datasets with respect to phylogenetic signal are only heightened by the spareness of the arachnid fossil record. Putting the informativeness of morphological characters aside, the appearances of chelicerate orders cannot be interpreted as a sequence of divergences wherein arachnids are unambiguously at the younger end of the chelicerate stratigraphic range (implying a later origin than aquatic chelicerate orders), because the chelicerate fossil record is prohibitively incomplete.

As a point of comparison, the fossil record of Pancrustacea is consistent with the sequence of divergence wherein hexapods are nested within the "crustaceans"—the stratigraphic ranges of the marine pancrustaceans extend into the Cambrian, whereas Hexapoda

do not appear until the Devonian. Within Hexapoda in turn, derived insect orders appear thereafter, in a window between the Jurassic and the Carboniferous. The temporal sequence of fossil appearances thus corroborates patterns predicted by molecular phylogenies of Pancrustacea (i.e., a series of nested relationships).

In Chelicerata, the oldest fossil horseshoe crabs (Xiphosura as well as synziphosurines) span the Ordovician in age (ca. 445–485 Myr old), whereas most fossil horseshoe crab taxa post-date the Ordovician-Silurian boundary [122]. The oldest unambiguous scorpion fossils (e.g., *Eramoscorpius brucensis*) are approximately 435 Myr old in age [123], occurring contemporaneously with Eurypterida (many of which postdate Silurian arachnid fossils). If scorpions are derived within Arachnopulmonata as the sister group of pseudoscorpions (again, an unambiguous split based on both rare genomic changes and phylogenomics [10,23]), then this phylogenomic placement, together with the age of *E. brucensis*, effectively guarantees that diversification of Arachnopulmonata must have predated 435 Myr. The age of *E. brucensis* cannot be taken as reflective of, much less synonymous with, the initial diversification of the arachnids—the group must be much older. Furthermore, the absences of other arachnopulmonate stem-group lineages (e.g., stem-Tetrapulmonata; stem-Pseudoscorpiones), as well as the apulmonate arachnid orders in Silurian strata only underscore the marked incompleteness of the chelicerate fossil record. Ordovician or older divergences of the arachnid assemblage would be more consistent with the derived placement of scorpions and molecular dating estimates of the basal arachnid diversification (regardless of the recovery of arachnid monophyly [18,116]) and would also accord with the estimated appearance of land plants by the Cambrian [54,124].

Thus, the temporal sequence of appearances of arachnid orders in the fossil record does not unambiguously substantiate inferences of arachnid monophyly. The distribution of chelicerate fossil ages is not consistent with the traditional interpretation of a gradual, stepwise colonization of land by a grade of merostomates, the scenario favored to this day by paleontologists [6]. This near-simultaneous appearance of major terrestrial and aquatic chelicerate groups in the fossil record just as easily accords with the short internodes at the base of the euchelicerate diversification in molecular phylogenies, in contradiction of the scenario of gradual and stepwise evolution of eurypterids and arachnids after their divergences from the horseshoe crab assemblage.

Given both low phylogenetic signal in morphological datasets, as well as the incompleteness of the chelicerate fossil record, it is not unexpected that chelicerate paleontologists have never successfully resolved a stable phylogeny of Chelicerata, despite the ability to sample extinct lineages and character states that are off-limits to molecular datasets. Recent paleontological phylogenies of chelicerates exhibit marked dissimilarity of topologies across different families of data matrices and analytical approaches, with broad disagreement on the basally branching placement of scorpions; the monophyly of Euchelicerata; and the monophyly of Tetrapulmonata. The strict consensus of chelicerate morphological phylogenies published in the past five years (after reanalyzing one study to remove an a priori constraint for arachnid monophyly) constitutes a total polytomy (Figure 1). The only relationships that are somewhat consistently recovered by morphological phylogenies (e.g., a basally branching placement of scorpions at the base of Arachnida; Pseudoscorpiones + Solifugae) happen to be exactly those that have been wholly refuted by evidence based on phylogenomics and genome architecture [10,21–23]. There is no evidence that the addition of fossils rescues chelicerate phylogeny from topological instability at the base of Euchelicerata, nor that morphological phylogenetic signal is consistent and unambiguous across datasets.

Given the topological uncertainty surrounding chelicerate relationships, molecular dating—an oft-cited justification for valuing morphological datasets—for the group may be a premature, error-prone exercise. Recent efforts to date the chelicerate tree of life have been based upon paralogy-riddled datasets [18,125], one of which went as far as estimating rates of cladogenesis on a phylogeny that undersampled the higher-level diversity of diverse groups of arachnids (Acariformes and Parasitiformes) and did not include all

extant arachnid orders [125]. These analyses provide little to testing the hypotheses of chelicerate relationships and evolution; the ability to place precise confidence intervals on the ages of unstable nodes that may not exist is not synonymous with evolutionary insight. In a group that bears over 130,000 described species, measures of diversification dynamics using a dataset of less than 100 exemplars that omit much of the higher-level ordinal diversity are meaningless. If the vertebrate literature serves as any guide, robust hypothesis-testing using diversification rate analyses and comparative methods requires many hundreds of terminals with dense (ideally, complete) species-level sampling. The authors of these premature analyses of chelicerate diversification dynamics [125] have repeatedly mistaken the ability to perform an analysis and obtain a palatable result for the notion of achieving "consilience". What they call consilience is more akin to stacking suppositions on top of other suppositions.

Consilience, strictly defined, refers to independent lines of scientific evidence arriving at the same conclusion. As explained above, independent lines of evidence (e.g., phylogenomic analyses; rare genomic changes; gene expression patterns) have repeatedly rejected the validity of several relationships postulated by morphological data partitions. The robust support for groups like Arachnopulmonata and Panscorpiones from independent lines of phylogenomic, developmental genetic, and genomic analyses underscore one of the most valuable principles of the scientific enterprise: validation.

In the context of phylogenetics, independently evolving partitions in molecular datasets can be identified and compared for validation of phylogenetic signal; genes and sites can be examined and subsampled to assess sources of systematic bias; whole genomes can be sequenced to search for rare genomic changes like microRNAs and transposon insertions; internal and external anatomical character systems can be surveyed and scrutinized for evidence of evolutionary history. In principle, for the case of ambiguous morphological characters, approaches in functional genetics and comparative development (evo-devo) can be used to understand the developmental genetic basis for trait convergence and thereby evaluate the validity of homology statements. The collection, comparison, and evaluation of all these data sources that substantiate or refute hypotheses of evolutionary relationships is integral to validation and the achievement of consilience. In Chelicerata, it was through such integration of different datasets that a robust phylogeny of enigmatic groups like sea spiders was recently obtained and competing hypotheses for the placement of the long-branch taxon Pseudoscorpiones were arbitered with rare genomic changes [10,100].

The ability to perform such validation through diverse and independent sources of phylogenetic data, however, is the domain of living organisms. With the exception of recent fossil taxa that can be sequenced (e.g., Neanderthals; moas; mammoths [126–128]) and applications of metazoan biomarkers for the interpretation of fossils (e.g., [129,130]), the study of paleontological systematics usually features access only to a single data class: morphology. As discussed above, the collection, interpretation, and codification of morphological data is fraught with subjectivity and inherent limitations, even for extant lineages. Tree topologies for the internal relationships of ancient extinct groups (e.g., Eurypterida; Synziphosurina) effectively lack validation by other data classes altogether. While methods exist to address homoplasy in morphological datasets (e.g., implied weighting), these congruence-based approaches rely upon the same morphological dataset that is being evaluated, and their interpretation is often subjective or circular.

There is usually no way for the practicing paleontologist to validate the relationships of fossils using data sources other than morphology. While this is not a barrier for the study of diverse assemblages with strong phylogenetic signal and large sample sizes (perhaps best exemplified by the and shelled mollusk and Mesozoic vertebrate fossil record), it represents a fundamental epistemological limitation of paleontological systematics for groups like chelicerates. When the fossil record is sparse and the parts prone to fossilization lack strong phylogenetic signal, how does the systematic paleontologist distinguish signal from investigator bias? How do they test the accuracy of the sole data class available to them?

And what happens if the systematic paleontologist has (as in the case of Chelicerata) a priori reason to doubt the informativeness of that sole available data class?

For neontologists, the decision would be simple: Abandon morphology, or alternatively, retain only the morphological character systems that clearly exhibit strong phylogenetic signal (by whatever criterion signal is evaluated [60,86,115]). Similar decisions are routinely made by molecular phylogeneticists for data partitions that are uninformative, noisy, data-poor, or low-quality—neontologists have access to various data classes and types, and thus typically place little investment in specific data partitions. Chelicerate paleontologists do not have this option—for some, admitting that arachnids may not be monophyletic is tantamount to conceding that their only available data class is unreliable. The downstream implication is that placements of other Paleozoic fossil groups could be little more than untestable conjecture.

As far as we can tell, the only communities that benefit from the perpetuation of arachnid monophyly are those with a vested interest in the validity of morphological phylogenies—principally, paleontologists and "molecular paleobiologists". Considering how much time, training, and investment has been poured into a specific scenario of chelicerate terrestrialization and morphological evolution over the past decades by adherents of these approaches, much is at stake for this traditionalist community when the nearly monolithic interpretation of arachnid terrestrialization is questioned. More generally, entrenched resistance to alternative evolutionary scenarios and hypotheses of character state transformations is a historically recurring phenomenon among some groups of morphologists and paleontologists, as exemplified by the history of Ecdysozoa and Pancrustacea.

It is for this reason that "molecular paleobiological" works like those of Lozano-Fernández et al. [42] and Howard et al. [18], which grasp for the defense of traditional relationships, do not represent balanced investigations grounded in consilience. They represent an advocacy, one that aims to defend a particular discipline, overestimates informativeness and completeness of a preferred data class, and overvalues a preferred set of preconceived hypotheses, in contradiction of the actual data, the sum of analyses, and the literature. Such an advocacy is irrevocably tainted with investigator bias, inasmuch as it is unwilling a priori to consider the conclusion that a preferred data class (e.g., morphology; the fossil record) could be flawed, uninformative, or incomplete, despite the weight of empirical evidence.

As is often the case when molecular data overturn long-held morphological hypotheses of relationships, adherents of Arachnida (principally, some morphologists and paleontologists) will likely continue to mount a spirited, but ultimately fruitless, effort to defend their preconceived vision of chelicerate phylogeny. As underscored by our previous works, accruing data from genome-scale analyses with comprehensive taxonomic sampling, site heterogeneous models, filtering for systematic biases, and total evidence approaches (that is, concomitant application of every approach embraced by "molecular paleobiologists") are overturning arachnid monophyly, the notion that arachnid paraphyly is attributable to an artifact, and an array of relationships previously thought to be robustly supported by morphological data (e.g., scorpions sister group to the remaining arachnids; pseudoscorpions + solifuges). As shown above, Arachnida has never been robustly supported even by morphological data.

Given that there are more characters supporting spurious, debunked relationships than those supporting arachnid monophyly, both in molecular datasets (e.g., Dromopoda) and in morphological ones (e.g., Atelocerata), we simply have to ask: What value is there in continuing to mainstream the delusion that arachnid monophyly is a certainty?

## 8. Last One Out, Get the Lights: The Future of Chelicerate Phylogeny

"Articulata . . . is based on much better morphological evidence [than] Ecdysozoa."

—Wägele and Misof (2001) [131]

" . . . morphological evidence suggests that myriapods are the sister group to Hexapoda."

—Wägele and Kück (2014) [132]

"I agree with you that the morphological support for Atelocerata [the debunked grouping of Hexapoda + Myriapoda] is stronger than that for Arachnida."

—Gregory D. Edgecombe (9 October 2019, personal communication to PPS)

In the interval since we first encountered the parable of the toucans, we have collectively engaged in a broad array of scientific approaches, such as taxonomic description and revision (including the description of fossils), bioinformatics, morphological and molecular phylogenetics, population genetics, phylogenomics, molecular dating, comparative genomics, developmental genetics, and evolutionary developmental biology (note, however, that we do not identify as "molecular paleobiologists", and we take serious exception to being labeled as such). While we recognize that every discipline bears its share of in-built assumptions and logical bridges, of all the approaches and methods we have pursued, the coding of morphological character matrices has been the most subjective, assumption-riddled, and validation-free activity we have engaged in, particularly as it impacts the placement of fossils. The characters and states we have encountered in chelicerate morphological datasets (even for extant taxa) are steeped in questionable assumptions of homology and artificial discretization of phenotypic traits, respectively. Given that the developmental genetic basis for the majority of arachnid morphological traits remains unknown, there is little validation available for the majority of traits that putatively inform chelicerate phylogeny. While we do not doubt that robust morphological matrices and sufficiently detailed fossil histories, with internal validation, are available for better test cases of the total evidence approach (e.g., echinoderms; shelled mollusks; vertebrates), we submit that chelicerates are simply and clearly not among this group.

Morphology has essentially outlived its usefulness as a phylogenetic data class for the higher-level relationships of numerous taxa that do not exhibit a large number of morphological features to score, lack detailed fossil records, exhibit a propensity for morphological convergence, or lack clear phylogenetic signal in anatomical character systems (e.g., Annelida; Nematoda; Sipuncula; Platyhelminthes; Cycliophora; Gnathostomulida; Chaetognatha). Despite the exhortations of a few adherents of the total evidence approach to collect more morphological data for combined analyses—even when molecular data have superiorly resolved the phylogenetic questions at hand in the same study (e.g., [60])—the notion that the next generation of evolutionary biologists will turn away in droves from fields like genomics and molecular evolution to dedicate time for coding morphological matrices is not realistic. It is detrimental to the next generation of arthropod morphologists to advertise the continuation of questionable phylogenetic practices, as well as to pass on the burden of defending unsupported, century-old hypotheses, as a viable course of scholarly pursuit for the future—particularly given that new finds and advanced imaging techniques have actively transformed our understanding of arthropod paleontology, outdated homology statements, and the evolution of character states in the past decade alone [63,96,133–135]. Paleontology has the potential to contribute greatly to the understanding of chelicerate evolution, but that potential is a function of how objectively fossil data and competing evolutionary scenarios are interpreted. It is far better to approach new discoveries in paleontology and morphology with an open mind than to cudgel their interpretations to accord with obsolete ideas and ossified biases.

Bluntly stated, morphology has nothing left to contribute to the defense of Arachnida, and unbiased analyses of molecular data outright refute it. Morphologists have had centuries to uncover characters for chelicerate relationships. We long ago hit the point of diminishing returns with regard to new character systems to score. Recent exceptions that have leveraged high-end imaging approaches to survey internal characters are certainly uncovering new character systems [136–139], but these only reinforce support for new hypotheses of relationships derived from phylogenomics, thereby further undermining morphological topologies that continue to be produced by chelicerate paleontologists (e.g., [5,7,9,89,91]). At some point, adherents of Arachnida need to weigh objectively whether their goal is to test a systematic hypothesis using unbiased assessments

of available evidence and balanced treatment of new ideas, or simply to affirm a traditional worldview—an affirmation that seems to require cherry-picking of taxa, matrices, characters, and substitution models, in tandem with shallow analytical interrogations of error-riddled datasets.

As was the case for proponents of Articulata and Atelocerata, adherents of Arachnida will gradually dwindle in number as their case inexorably erodes. The field will continue to move away from the limitations of subjective character interpretations and untested assumptions passed down through time, leaving Arachnida behind in the dustbin of obsolete morphology-based groupings. After a time, the post hoc realignment of morphological interpretations to accord with the new outcomes of molecular phylogenies will bury an older generation of paleontology- and morphology-driven hypotheses of chelicerate relationships, paralleling the history of Ecdysozoa and Pancrustacea [81,91]. In the case of Arachnopulmonata and Panscorpiones, this process is already well underway [10,137]. Inevitably, when this post hoc realignment is completed (usually, about a decade after molecular datasets have cracked the case), some paleontologists and "molecular paleobiologists" will proclaim its success as a victory and a vindication of morphology, without a trace of irony (e.g., [91]).

Propitious developments in chelicerate genomics foretell a bright future for the study of chelicerate phylogeny and evolution. The discovery of rare genomic changes in the chelicerate tree of life has offered a new and powerful class of data for testing competing tree topologies and validating inferential approaches to phylogenomics—enabling arachnologists to break century-old impasses in the placement of groups like scorpions and pseudoscorpions [10,20]. The advent of "molecular morphology" in the form of analyses of genomic architecture, together with independent evaluations of various data classes (e.g., gene expression patterns; microRNA incidence and enrichment; gene family duplications; [10,20–27]), offers a robust vehicle for the achievement of consilience in chelicerate evolutionary relationships. The parallel trend of accruing functional genetic tools for various chelicerate model systems offers new routes to understanding morphological traits and their evolution, as well as discovering functional links between genomes and phenotypes [20,29,34,140–145]. It is in such vehicles that future scholars of chelicerate evolution should direct their efforts, energy, and attention.

**Author Contributions:** Conceptualization, P.P.S., J.A.B. and C.E.S.-L.; methodology and analysis, P.P.S. and J.A.B.; writing—original draft preparation, P.P.S.; writing—review and editing, J.A.B. and C.E.S.-L.; visualization, P.P.S.; funding acquisition, P.P.S. All authors have read and agreed to the published version of the manuscript.

**Funding:** This research was funded by the National Science Foundation, grant numbers IOS-1552610 and IOS-2016141. The APC was funded by IOS-2016141.

**Data Availability Statement:** Datasets corresponding to Figures 3–5 are available on FigShare: https://figshare.com/articles/dataset/Supplementary_material_for_Sharma_et_al_2021_-_Diversity/16913386.

**Acknowledgments:** We are indebted to Matjaž Kuntner for the invitation to contribute to this special issue. Gonzalo Giribet, Georg Brenneis, and Claudia P. Arango provided photographs in Figure 6. Comments from Matjaž Kuntner and two anonymous reviewers improved a previous draft of the work.

**Conflicts of Interest:** The authors declare no conflict of interest. The funders had no role in the design of the study; in the collection, analyses, or interpretation of data; in the writing of the manuscript, or in the decision to publish the results.

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
