# Peer review of "What Is an “Arachnid”? Consensus, Consilience, and Confirmation Bias in the Phylogenetics of Chelicerata"

_diversity, doi:10.3390/d13110568_

Round 1

Reviewer 1 Report

This paper provides a review of the monophyly (or, better said, lack thereof) of Arachnida based on previously published datasets of either morphological or phylogenomic data. Although new analyses are not presented and the discussion is mostly discursive, a strong point is made towards the non-monophyly of Arachnida. Perhaps it would be elegant to provide re-analyses of morphological matrices in those cases where it is argued that previous scorings were incorrect, although this is not quite necessary.

I appreciate that the authors strive to write in an elegant style, but as a non-native speaker it is mildly annoying to have to stop reading the paper every now and then to look in a thesaurus for non-usual words (e.g. “buttressed”, “apropos”, “ancillary”) which have plenty of more well-known synonyms. This hampers understanding the scientific content. I suggest the authors have this in mind and strive for clarity rather than style. Scientific writing is already exclusionary enough for people who have English as a secondary language.

In other portions, I find the tone to be excessively aggressive towards past research and think it could be toned down in some portions (outlined below). I also found some minor typos indicated below. Otherwise the paper is a nice read and a very good overview of this subject.

127: placemnet

153: “Due to the systemic nature of whole genome duplication events, the weight of evidence was decidedly in favor of phylogenomic results and contrary to morphological analyses (Figure 2c).”. To me, it is not self-evident why whole genome duplication events are better evidence than the alternatives to support a specific placement (especially considering they happened at least three times in the phylogeny in 2c). Please further develop this argument.

171: phylogentically

219-243: this paragraph is a bit hard to follow because there are different studies from different teams, in some cases it is not immediately clear who did what (e.g. 227, “While the same research team”… it is not readily clear which research team). Because the same matrices have been reused, rebuilt, or re-analyzed using different models, I suggest it might be better to try to summarize this story in a figure showing the pipeline of matrices / methods, indicating the research team and the phylogenetic outcome.

364: molcular

366-395: as you state in the beginning of the paragraph, this passage is completely parenthetical, and I do not see what it contributes to the main argument of the paper; in some portions it reads almost ad hominem as it does not provide concrete examples of the claims you are making. I do agree that the term “molecular paleobiologist” is misleading, and even ridiculous, when used in this context (I would think that a molecular paleobiologist is someone who studies fossilized molecules such as ancient proteins or DNA, not someone who is doing integrative systematics), so perhaps that discussion is valid, but it could be significantly shortened and toned down in my opinion.

582: recovery > recover

594-603: again, I think this part is a bit ad hominem and could be toned down – perhaps just emphasizing that published work without a publicly available dataset is less than desirable and precludes reproducibility.

614: arachnds

619-620: to play devil’s advocate, would scoring this as inapplicable to Solifugae, Acari and Palpigradi leading to disruption of Arachnida? I think that perhaps the paper would gain from a reanalysis of the same datasets with the corrected scorings

Reviewer 2 Report

The manuscript of Sharma and coauthors review the extensive debate generated about the Chelicerata phylogeny, most specifically the monophyly of Arachnida.  They reanalyzed data from other authors that recovered Arachnida monophyly with molecular data in order to explore problems of distinguishing noise from phylogenetic signal, questioning the approach and the results obtained in previous studies. Additionally, the authors disputed the morphological support of arachnid monophyly, which they argued, it is mainly recovered by the use of incorrect or ambiguous coded characters.

In my opinion it is a good review on which the authors explore through an extensive updated references the different hypothesis about the chelicerate relationships and arachnid monophyly. I have no doubt that this review will intensify the current debate about arachnid monophyly, and the use of preferred molecular data over traditional morphology in the case of groups where the fossil records are scarce to apply a total evidence approach (like is the case of chelicerates). But I think that the authors should down the tone in some paragraphs because their criticisms about the “practices” of other authors sometimes seem a bit aggressive (see below some examples).

L 298 I have not very clear how do the authors reconstruct the matrices and topologies. Maybe the authors could add more details about the in silico experiments they performed.

Some paragraphs could be reduced or synthetized a little bit. E.g. L 366, section 5 and 6.

Sentences or paragraphs where the authors could down the tone: E.g.

L 187. (and de facto unscientific) exercise

L 259. reflecting noise and bioinformatic incompetence, rather than phylogenetic signal.

L336. Figure 5 legend. (In other words, if you fell for this argument, articulated by Lozano-Fernández et al. [42] in support of arachnid monophyly, you got fooled.).

L379. in a certain light, these works represent a form of intellectual parasitism, in that they rely on datasets generated by other groups to exist.

L424. a  low-quality scholarship that effectively prevents the field from advancing past the gains of morphological cladistics in the 1980s. In the worst light, this practice represents a gateway to pseudoscience and possible fraud.

L589. that led us to conclude that the original matrices were lost, incorrectly analyzed, or possibly never generated as reported in their corresponding publications.

L594. the ramifications of publishing a tree with the wrong dataset, or with no dataset, on multiple occasions, and in peer-reviewed journals.

Minor issues:

L145 Figure 3 should be Figure 2 (as referred in the main text).

L127 misspelling placemnet

L171 misspelling phylogenetically

L303 misspelling construed as                

L364 misspelling “molcular

L614 misspelling all arachnds,

L806 peo-ple’s

L821 misspelling matgrices,
